# Frequency-aware Generative Models for Multivariate Time Series Imputation

**Xinyu Yang**[1], **Yu Sun**[1*], **Xiaojie Yuan**[1], **Xinyang Chen**[2*]
[1]College of Computer Science, DISSec, Nankai University, China
{yangxinyu@dbis.,sunyu@,yuanxj@}nankai.edu.cn
[2]School of Computer Science and Technology, Harbin Institute of Technology, Shenzhen, China
chenxinyang@hit.edu.cn

## Abstract

Missing data in multivariate time series are common issues that can affect the analysis and downstream applications. Although multivariate time series data generally consist of the trend, seasonal and residual terms, existing works mainly focus on optimizing the modeling for the first two items. However, we find that the residual term is more crucial for getting accurate fillings, since it is more related to the diverse changes of data and the biggest component of imputation errors. Therefore, in this study, we introduce frequency-domain information and design **F**requency-aware **G**enerative Models for Multivariate **T**ime Series **I**mputation (**FGTI**). Specifically, FGTI employs a high-frequency filter to boost the residual term imputation, supplemented by a dominant-frequency filter for the trend and seasonal imputation. Cross-domain representation learning module then fuses frequency-domain insights with deep representations. Experiments over various datasets with real-world missing values show that FGTI achieves superiority in both data imputation and downstream applications.

## 1 Introduction

Missing data are commonly observed in the multivariate time series due to diverse reasons [25], which would encumber subsequent analysis and applications [17]. It is not surprising that more accurate missing data imputation generally leads to better performance in downstream applications.[1]

Existing techniques [13; 28] have revealed that time series data can be decomposed into three distinct terms, i.e., trend, seasonal, and residual, and try to compute an imputation by modeling the first two items as accurately as possible. Figure 1 reports a survey for the imputation accuracy of the three terms by representative imputation methods over the pre-decomposed KDD [6] dataset with 10% missing values.[2] The dataset comprises 8,034 consecutive readings of meteorological and air quality data taken over a year in Beijing. In this dataset, the trend term may reflect long-term changes in climate or air quality conditions, and the seasonal term might capture patterns associated with different seasons. Moreover, the residual term may consist of short-term, irregular, and high-frequency changes. As shown in Figure 1, the imputation error is mainly caused by the residual term, which has not been well studied, unfortunately.

Recent studies indicate that the high-frequency components are intricately related to the residual [46; 58; 26] and contain critical information for imputing the residual term. Unfortunately, deep learning architectures cannot generalize well in modeling high-frequency components. [38; 43]. To

---

[*]Corresponding authors.
[1]Please see an empirical study in Section 4.5.
[2]Please see Section 2 for a brief survey, see Appendix A.5.1 for detailed experiment setup.

38th Conference on Neural Information Processing Systems (NeurIPS 2024).

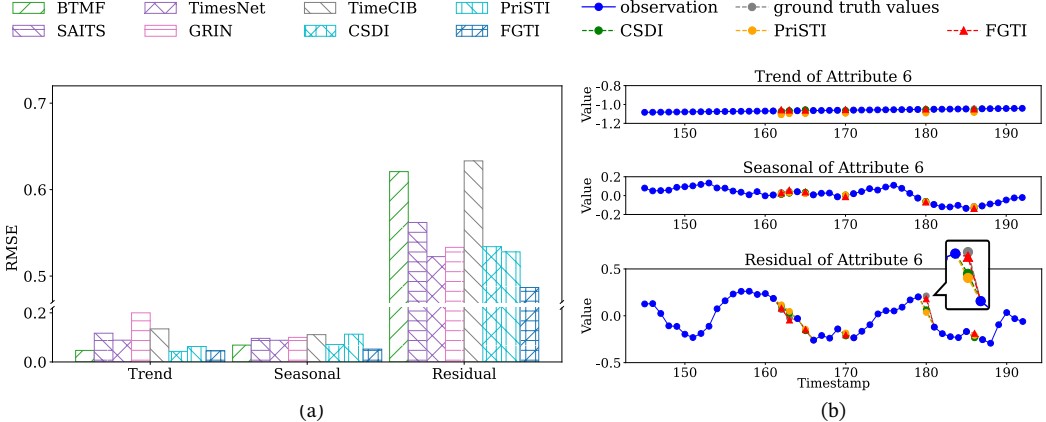

Figure 1: Improving the imputation accuracy of the residual term is the key to boosting the imputation performance of the model.

meet this challenge, we design **F**requency-aware **G**enerative Models for Multivariate **T**ime Series **I**mputation (**FGTI**), which can extract frequency-domain information and use two cross-domain (i.e., time-domain and frequency-domain) representation learning modules to guide the generation process of deep models. Specifically, we start with the high-frequency filter designed to extract the high-frequency information essential for guiding the accurate imputation of the residual term. This choice is consistent with recognizing the critical role of high-frequency information in imputation accuracy. Then, we introduce the dominant-frequency filter to address the potential challenge posed by high-frequency information for imputing trend and seasonal terms [4]. Furthermore, our cross-domain representation learning frameworks combine frequency-domain information with deep representations in the time-domain, enabling seamlessly intertwining frequency-domain information with time and attribute dependencies modeling.

Our research makes several notable contributions:

- We design a frequency-aware generative model FGTI with frequency-domain information integrated by the high-frequency filter and the dominant-frequency filter, to enhance the awareness of the frequency-domain.

- We introduce two cross-domain representation learning modules that provide models with prior knowledge of intricate frequency-related patterns for missing data imputation.

- We evaluate FGTI on three time series datasets with real-world missing values, which demonstrates the superiority of FGTI in both imputation accuracy and downstream applications.

## 2 Related work

Traditional imputation methods usually employ the statistics, such as mean value [23], median value [16], or last observed value [3], to impute missing data for multivariate time series. It is not surprising that such traditional signals cannot make full use of the valuable semantics of available data. BTMF [9] and TIDER [28] employ the low-rank matrix factorization to impute missing data. Unfortunately, due to the matrix capacity's limitation, it is still challenging to accurately match imputation values with the underlying complex relationships and dependencies.

Many studies have shown that deep learning based imputation methods are effective to fill multivariate time series data, such as BRITS [6], TST [57], SAITS [14], STCPA [54], TimesNet [52]. According to [48], forecasting models [50; 55] can also be applied to imputation task. Additionally, GRIN [12] and DAMR [39] use graph neural networks to incorporate known relationships between attributes. However, these methods with fixed model outputs cannot capture the uncertainty and variability of missing values. Recently, researchers have attempted to utilize large language models (LLMs) as the backbone for time series analysis [59]. Since there are significant differences between time series data and natural language, it still has a lot of room for improvement.

To capture the uncertainty and variability, researchers introduce generative models [8] into the missing data imputation by learning the implicit distribution of missing values. In the early stage, researchers mainly use variational Autoencoders (VAE) or generative adversarial networks (GAN) to generate new samples that match the distribution of the training dataset and impute missing values with the generated samples. VAE-based methods learn data distributions by optimizing the reconstruction error and regularizing in the latent space [31; 15; 35; 36; 11]. However, they may not capture the complex variability of missing data well and may produce inaccurate results, especially when the latent spaces are not well aligned. GAN-based approaches use the adversarial training technique of the generator and discriminator to improve the imputation results [56; 29; 30; 34]. Unfortunately, they may face convergence difficulties that can affect the imputation accuracy.

Diffusion models have been introduced into the imputation task recently, considering the success in various fields [40; 18; 24]. To impute time series data, CSDI [44] designs conditional score-based diffusion models with the conditions only on observed values, and SSSD [2] utilizes both conditional diffusion models and structured state space models. Additionally, MIDM [49] develops the noise sampling, addition, and denoising mechanisms, and PriSTI [27] further studies the enhanced prior modeling by extracting spatio-temporal dependencies as contextual conditions for spatio-temporal data imputation. However, as analyzed in the introduction, existing studies underestimate the importance of accurately modeling the residual term for missing data imputation.

For the time series imputation methods in frequency domain, mvLSWimpute [51] utilizes wavelet transforms to guide imputation, APDNet [60] uses the Fourier Temporal and Fourier Variable Interaction modules to model dependencies. In addition, the frequency domain time series forecasting methods FEDformer [58], FreTS [55] can also be applied to imputation task. However, they did not consider using frequency domain information to model the missing data's residual terms accurately, which is critical for boosting the overall imputation performance.

In contrast, our FGTI captures high-frequency information and dominant-frequency information to get a more accurate modeling of the residual term, while assisting in describing trend and seasonal terms.

## 3 Frequency-aware Generative Models

In this paper, we focus on the incomplete multivariate time series imputation problem. The input multivariate time series $\mathbf{X} = (\mathbf{X}_1, \ldots, \mathbf{X}_D) \in \mathbb{R}^{D \times L}$ is a set of $D$ attribute values recorded at $L$ consecutive timestamps, where each attribute series $\mathbf{X}_d \in \mathbb{R}^L$. Each element $x_{ij}$ in $\mathbf{X}$ is the observation of the $i$-th attribute at the $j$-th timestamp, which is probably missing. We use the binary mask matrix $\mathbf{M} \in \{0, 1\}^{D \times L}$ to represent the missing status of observations in $\mathbf{X}$, where $m_{ij} = 1$ in $\mathbf{M}$ denotes that $x_{ij}$ is complete, otherwise $x_{ij} = 0$. In our context, we refer to the imputation target as $\hat{\mathbf{X}}$. During the training of the imputation model, we choose some observations as the imputation target. When we impute missing values, we treat all the missing values as the imputation target.

In Figure 1, it is evident that the main obstacle to improving multivariate time series imputation is the residual term. Considering a recognized fact that the residual term often contains high-frequency components from the perspective of Fourier analysis [46; 58; 26], introducing prior knowledge of frequency-domain information can be a feasible approach to enhancing model performance.

As a class of superior data imputation models, deep generative models treat the time series imputation task as calculating the conditional imputation target probability distribution $q(\hat{\mathbf{X}}^0 | \mathbf{C})$, where $q(\hat{\mathbf{X}}^0)$ is the clean data distribution and existing deep generative imputation models [56; 29; 44] use the observed values $\mathbf{X}$ in the time-domain as the condition $\mathbf{C}$ for probability distribution calculation. Note that we denote the complete or the imputed imputation target as the clean imputation target $\hat{\mathbf{X}}^0$.

However, frequency principal [38; 43] reveals that deep models cannot generalize well to high-frequency information. As a result, it may not accurately impute the residual term [13] inherent in the time series dataset that cannot be trivialized in the imputation task [28].

To tackle the above challenge, we incorporate frequency-domain information into condition $\mathbf{C}$ to enhance the performance of generative models. Our FGTI implements the condition $\mathbf{C}$ that contains time-domain observation condition $\mathbf{X}^\mathbf{C}$, as well as the frequency-domain conditions $\mathbf{C}^\mathbf{H}$ and $\mathbf{C}^\mathbf{D}$.

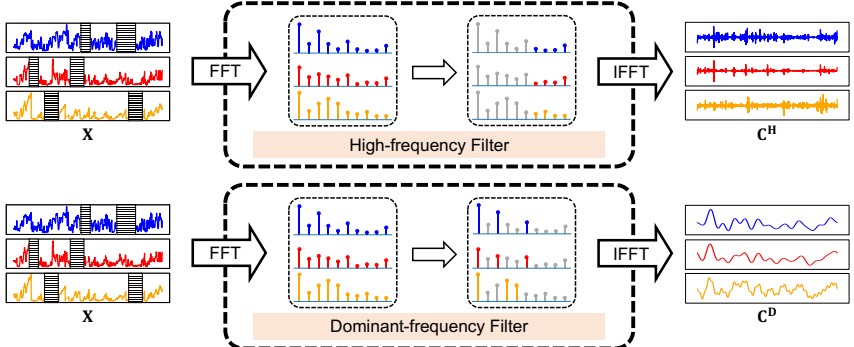

Figure 2: The high-frequency filter with $\mathcal{F} = 0.3$ and the dominant-frequency filter with $\kappa = 3$.

## 3.1 Frequency-domain Condition Filter

Our frequency-domain condition includes the nonlinear transformation of two parts: the high-frequency condition that guides the residual term and the dominant-frequency condition that contains background structure information to impute the trend and seasonal terms, as shown in Figure 2.

### 3.1.1 High-frequency Filter

The high-frequency filter extracts high-frequency information $\mathbf{C^H}$ from the time-domain observations, which is used to guide the imputation of residual terms in time series. Since different attributes in a multivariate time series can be heterogeneous, we consider extracting high-frequency information separately for each attribute series $\mathbf{X}_d \in \mathbb{R}^L$, where $d = 1, \ldots D$.

We first interpolate $\mathbf{X}_d$, then obtain the amplitude vector $\mathbf{A} \in \mathbb{R}^{\lfloor (L+1)/2 \rfloor}$ for $\mathbf{X}_d$ over sample frequency components $\mathbf{F} = \{\frac{1}{L}, \ldots, \frac{1}{L}\lfloor (L+1)/2 \rfloor\}$ by the Fast Fourier Transform (FFT):

$$(\mathbf{A}, \mathbf{F}) = \text{FFT}(\mathbf{X}_d). \tag{1}$$

To get the high-frequency condition, we discard the frequency components below a cutoff threshold $\mathcal{F}$ and map the remaining components to time-domain by the Inverse Fast Fourier Transform (IFFT),

$$\mathbf{C}_d^{\mathbf{H}} = \text{IFFT}\left[\mathbf{A} \odot (\mathbf{F} > \mathcal{F})\right], \tag{2}$$

where $\odot$ denotes the Hadamard product.

Finally, we concatenate the corresponding high-frequency information vectors $\mathbf{C}_d^{\mathbf{H}}$ for each attribute sequence to form the high-frequency condition $\mathbf{C^H} \in \mathbb{R}^{D \times L}$,

$$\mathbf{C^H} = \text{Concat}\left(\left\{\mathbf{C}_d^{\mathbf{H}}\right\}_{d=1}^{D}\right). \tag{3}$$

For the whole input time series $\mathbf{X}$, the time complexity of performing the high-frequency filtering is $\mathcal{O}(DL \log L)$. Since the time complexity of performing FFT for each attribute series is $\mathcal{O}(L \log L)$, selecting high-frequency components in the frequency domain has a time complexity of $\mathcal{O}(L)$, and performing IFFT costs $\mathcal{O}(L \log L)$ time.

### 3.1.2 Dominant-frequency Filter

The conditions extracted by the dominant-frequency filter from the time-domain observations not only provide the background structure information for generative models to guide the imputation of the trend and seasonal terms, but also mitigate the interference of the high-frequency condition on the imputation of the trend and seasonal terms[3].

The dominant-frequency information is mainly composed of frequency components with large amplitudes. If we have obtained the representation $(\mathbf{A}, \mathbf{F})$ of $\mathbf{X}_d$ in the frequency-domain from Equation 1, we can find the top-$\kappa$ frequency components with the largest amplitude according to $\mathbf{A}$,

$$\{f_1, \ldots, f_\kappa\} = \arg_{\mathbf{F}} \text{top}\kappa(\mathbf{A}). \tag{4}$$

---

[3]Please see an empirical study in Section A.5.1.

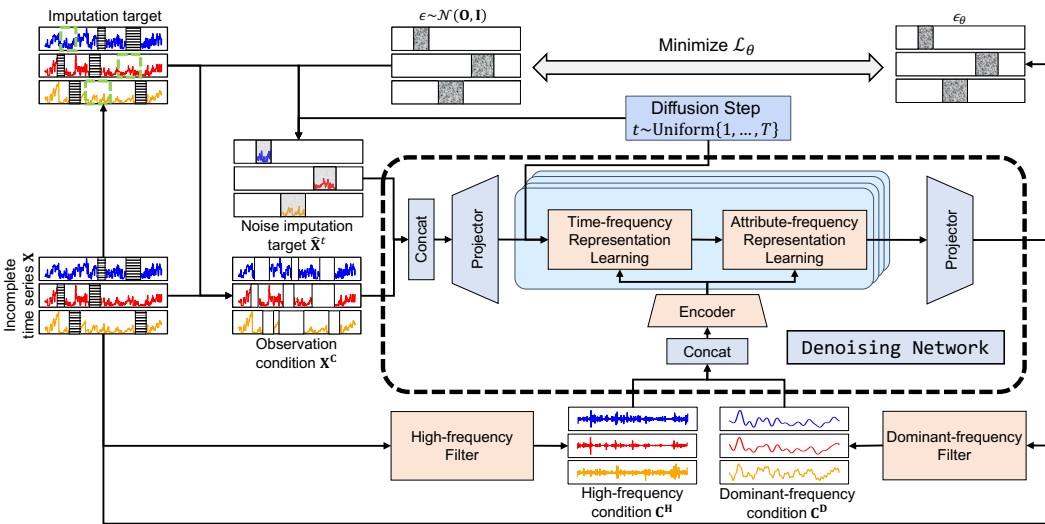

Figure 3: The pipeline of FGTI implemented by the frequency-aware diffusion model. FGTI incorporates high-frequency representations to guide the residual term and compensates for the trend and seasonal terms with the dominant-frequency representations. With cross-domain representation learning, our FGTI includes frequency-domain information into time and attribute dependencies modeling to estimate the diffusion noise.

Then we project these $\kappa$ frequencies to the time-domain via the Inverse Fast Fourier Transform,

$$\mathbf{C}_d^{\mathbf{D}} = \text{IFFT}\left[\mathbf{A} \odot \left(\mathbf{F} \in \{f_1, \ldots, f_\kappa\}\right)\right]. \tag{5}$$

Finally, we compose the dominant-frequency condition $\mathbf{C}^{\mathbf{D}} \in \mathbb{R}^{D \times L}$ by concatenating all $\mathbf{C}_d^{\mathbf{D}}$,

$$\mathbf{C}^{\mathbf{D}} = \text{Concat}\left(\left\{\mathbf{C}_d^{\mathbf{D}}\right\}_{d=1}^D\right). \tag{6}$$

Similar to the high-frequency filter, the time complexity of performing dominant-frequency filtering for the whole input $\mathbf{X}$ is $\mathcal{O}(DL \log L)$.

## 3.2 Cross-domain Representation Learning

With the aim of integrating frequency-domain conditions into deep generative models, we first use an encoder to map the conditions to representation $\mathbf{C}^{\mathbf{F}} \in \mathbb{R}^{D \times L \times K}$ in the latent space,

$$\mathbf{C}^{\mathbf{F}} = \text{Encoder}\left[\text{Concat}\left(\mathbf{C}^{\mathbf{H}}, \mathbf{C}^{\mathbf{D}}\right)\right], \tag{7}$$

where $K$ is the channel number of the latent space. In this paper, we implement the $\text{Encoder}(\cdot)$ with the well-acknowledged transformer [47] backbone, since it can self-adaptively extract critical information in $\mathbf{C}^{\mathbf{H}}$ and $\mathbf{C}^{\mathbf{D}}$ by the self-attention mechanism.

To accurately capture time and attribute dependencies guided by frequency-domain information, we design two frameworks: Time-frequency representation learning and Attribute-frequency representation learning. They integrate the current intermediate hidden representations $\mathbf{R}^{in} \in \mathbb{R}^{D \times L \times K}$ in the time-domain of deep generative models with the frequency-domain representation $\mathbf{C}^{\mathbf{F}}$.

Specifically, we use the cross-attention mechanism that can efficiently learn the various input modalities [21; 40] for representation fusion.

### 3.2.1 Time-frequency Representation Learning

To capture time dependencies with the aid of frequency-domain information, we divide input hidden representation $\mathbf{R}^{in} = \{\mathbf{R}_d^{in} \in \mathbb{R}^{L \times K}\}_{d=1}^D$ and frequency information $\mathbf{C}^{\mathbf{F}} = \{\mathbf{C}_d^{\mathbf{F}} \in \mathbb{R}^{L \times K}\}_{d=1}^D$ into $D$ segments according to attributes. For each pair of latent representation segment $(\mathbf{R}_d^{in}, \mathbf{C}_d^{\mathbf{F}})$,

the learning process to obtain time-frequency representation of each attribute $\mathbf{R}_d^{\mathsf{t}} \in \mathbb{R}^{L \times K}$ is

$$\mathbf{R}_d^{\mathsf{t}} = \mathrm{Softmax}\left(\frac{\mathbf{Q}_d \mathbf{K}_d^{\top}}{\sqrt{K}}\right) \cdot \mathbf{V}_d, \tag{8}$$

where $\mathbf{Q}_d = \mathbf{C}_d^{\mathbf{F}} \cdot \mathbf{W}_{\mathsf{t}}^{\mathbf{Q}}$, $\mathbf{K}_d = \mathbf{C}_d^{\mathbf{F}} \cdot \mathbf{W}_{\mathsf{t}}^{\mathbf{K}}$, $\mathbf{V}_d = \mathbf{R}_d^{in} \cdot \mathbf{W}_{\mathsf{t}}^{\mathbf{V}}$, $\mathbf{W}_{\mathsf{t}}^{\mathbf{Q}}, \mathbf{W}_{\mathsf{t}}^{\mathbf{K}}, \mathbf{W}_{\mathsf{t}}^{\mathbf{V}} \in \mathbb{R}^{K \times K}$ are learnable weight matrices.

Then we concatenate $\mathbf{R}_l^{\mathsf{t}}$ of all attributes to obtain the representation $\mathbf{R}^{\mathsf{t}} \in \mathbb{R}^{D \times L \times K}$,

$$\mathbf{R}^{\mathsf{t}} = \mathrm{Concat}\left(\{\mathbf{R}_d^{\mathsf{t}}\}_{d=1}^{D}\right). \tag{9}$$

### 3.2.2   Attribute-frequency Representation Learning

To capture dependencies between different attributes based on frequency-domain information, we divide the latent time-frequency representation $\mathbf{R}^{\mathsf{t}} = \{\mathbf{R}_l^{\mathsf{t}} \in \mathbb{R}^{D \times K}\}_{l=1}^{L}$ and $\mathbf{C}^{\mathbf{F}} = \{\mathbf{C}_l^{\mathbf{F}} \in \mathbb{R}^{D \times K}\}_{l=1}^{L}$ into $L$ segments, according to timestamps. The learning process to obtain attribute-frequency representation of each timestamp $\mathbf{R}_l^{\mathfrak{a}} \in \mathbb{R}^{L \times K}$ is as follows:

$$\mathbf{R}_l^{\mathfrak{a}} = \mathrm{Softmax}\left(\frac{\mathbf{Q}_l \mathbf{K}_l^{\top}}{\sqrt{K}}\right) \cdot \mathbf{V}_l, \tag{10}$$

where $\mathbf{Q}_l = \mathbf{C}_l^{\mathbf{F}} \cdot \mathbf{W}_{\mathfrak{a}}^{\mathbf{Q}}$, $\mathbf{K}_l = \mathbf{C}_l^{\mathbf{F}} \cdot \mathbf{W}_{\mathfrak{a}}^{\mathbf{K}}$, $\mathbf{V}_l = \mathbf{R}_l^{tf} \cdot \mathbf{W}_{\mathfrak{a}}^{\mathbf{V}}$. To get the updated representation $\mathbf{R}^{\mathfrak{a}}$, we need to concatenate all $\mathbf{R}_l^{\mathfrak{a}}$ according to timestamps,

$$\mathbf{R}^{\mathfrak{a}} = \mathrm{Concat}\left(\{\mathbf{R}_l^{\mathfrak{a}}\}_{l=1}^{L}\right). \tag{11}$$

### 3.3   Frequency-aware Diffusion Model

Recently, diffusion generative models have demonstrated remarkable proficiency and have emerged as the leading generative models in numerous fields [24; 18]. Thus, we take the diffusion model as an example to introduce how to use frequency-domain conditions to boost missing data imputation.

Specifically, we implement FGTI by the frequency-aware diffusion model. Our frequency-aware diffusion model fuses with frequency-domain conditions to learn the conditional imputation target distribution $q(\hat{\mathbf{X}}^0 \mid \mathbf{X}, \mathbf{C}^{\mathbf{H}}, \mathbf{C}^{\mathbf{D}})$, through two Markov chain processes of diffusion step $T$, i.e., the diffusion forward process and the diffusion reverse process.

The diffusion forward process involves gradually adding Gaussian noise into the imputation target,

$$q(\hat{\mathbf{X}}^{1:T} \mid \hat{\mathbf{X}}^0) = \prod_{t=1}^{T} q(\hat{\mathbf{X}}^t \mid \hat{\mathbf{X}}^{t-1}), \tag{12}$$

where $q(\hat{\mathbf{X}}^t \mid \hat{\mathbf{X}}^{t-1}) = \mathcal{N}(\sqrt{\alpha^t}\hat{\mathbf{X}}^{t-1}, \beta^t \mathbf{I})$, $\beta^t \in (0, 1)$, is a hyperparameter satisfying $\beta^t < \beta^{t+1}$ for $t = 1, \ldots, T-1$. In addition, $\alpha^t = 1 - \beta^t$ and $q(\hat{\mathbf{X}}^0)$ is the complete data distribution.

According to DDPM [20], $\hat{\mathbf{X}}^t$ has a closed-form solution,

$$\hat{\mathbf{X}}^t = \sqrt{\overline{\alpha^t}}\hat{\mathbf{X}}^0 + \sqrt{1 - \overline{\alpha^t}}\epsilon, \tag{13}$$

where $\epsilon \sim \mathcal{N}(\mathbf{0}, \mathbf{I})$, $\overline{\alpha^t} = \prod_{i=1}^{t} \alpha^i$. Therefore, we can directly obtain $\hat{\mathbf{X}}^t$ from $\hat{\mathbf{X}}^0$. The details for deriving the closed-form solution can be found in Appendix A.2.1. Note that when $T$ is large enough, $q(\hat{\mathbf{X}}^T \mid \hat{\mathbf{X}}^0) \approx q(\hat{\mathbf{X}}^T), q(\hat{\mathbf{X}}^T) \approx \mathcal{N}(\mathbf{0}, \mathbf{I})$.

Our diffusion reverse process gradually removes Gaussian noises added to the imputation target based on the time-domain observation condition $\mathbf{X}^{\mathbf{C}}$, the high-frequency condition $\mathbf{C}^{\mathbf{H}}$ and the dominant-frequency condition $\mathbf{C}^{\mathbf{D}}$, which can be formalized as the Markov chain,

$$p_{\theta}(\hat{\mathbf{X}}^{T-1:0} \mid \hat{\mathbf{X}}^T) = \prod_{t=1}^{T} p_{\theta}(\hat{\mathbf{X}}^{t-1} \mid \hat{\mathbf{X}}^t, \mathbf{X}^{\mathbf{C}}, \mathbf{C}^{\mathbf{H}}, \mathbf{C}^{\mathbf{D}}), \tag{14}$$

where $p_\theta(\hat{\mathbf{X}}^{t-1} \mid \hat{\mathbf{X}}^t, \mathbf{X}^\mathbf{C}, \mathbf{C^H}, \mathbf{C^D}) = \mathcal{N}\left(\mu_\theta\left[\hat{\mathbf{X}}^{t-1} \mid \hat{\mathbf{X}}^t, \mathbf{X}^\mathbf{C}, \mathbf{C^H}, \mathbf{C^D}\right], \left[\sigma^{t-1}\right]^2\right)$.

We next show that introducing the high-frequency condition $\mathbf{C^H}$ and dominant-frequency condition $\mathbf{C^D}$ can reduce the uncertainty of the diffusion reverse process, improving the imputation accuracy.

**Proposition 3.1.** *The conditional entropy*

$$\mathrm{H}\left(\hat{\mathbf{X}}^{t-1} \mid \hat{\mathbf{X}}^t, \mathbf{X}^\mathbf{C}, \mathbf{C^H}, \mathbf{C^D}\right) < \mathrm{H}\left(\hat{\mathbf{X}}^{t-1} \mid \hat{\mathbf{X}}^t, \mathbf{X}^\mathbf{C}\right),$$

*with additional high-frequency condition $\mathbf{C^H}$ and dominant-frequency condition $\mathbf{C^D}$ in the diffusion reverse process.*

It can be proved by the chain rule of the conditional entropy, as detailed in Appendix A.1.

Based on the classifier-free guidance diffusion model [19; 20], $p_\theta(\hat{\mathbf{X}}^{t-1} \mid \hat{\mathbf{X}}^t, \mathbf{X}^\mathbf{C}, \mathbf{C^H}, \mathbf{C^D})$ can be parameterized as $\mu_\theta\left[\hat{\mathbf{X}}^{t-1} \mid \hat{\mathbf{X}}^t, \mathbf{X}^\mathbf{C}, \mathbf{C^H}, \mathbf{C^D}\right] = \frac{1}{\sqrt{\alpha^t}}\left[\hat{\mathbf{X}}^t - \frac{\beta^t}{\sqrt{1-\overline{\alpha^t}}}\epsilon_\theta\left(t, \hat{\mathbf{X}}^t, \mathbf{X}^\mathbf{C}, \mathbf{C^H}, \mathbf{C^D}\right)\right]$, $\left[\sigma^{t-1}\right]^2 = \frac{(1-\overline{\alpha^{t-1}})\beta^t}{1-\overline{\alpha^t}}$, where $\epsilon_\theta(\cdot)$ is the denoising network with learnable parameter set $\theta$ as present in Appendix A.3.1. The mathematical details are presented in Appendix A.2.2.

As shown in Figure 3, the denoising network incorporates frequency-domain information into modeling time dependencies and attribute dependencies, through the time-frequency representation learning module and attribute-frequency representation learning module to guide the denoising.

For training the denoising network, we randomly select some observed values as the imputation target $\hat{\mathbf{X}}$ and use the remaining observations as the observation condition $\mathbf{X}^\mathbf{C}$ for each update step, since the ground truth of missing values is unknown. We train the denoising network by minimizing the following objective function $\mathcal{L}_\theta$,

$$\mathcal{L}_\theta = \mathbb{E}\left\|\epsilon - \epsilon_\theta\left(t, \hat{\mathbf{X}}^t, \mathbf{X}^\mathbf{C}, \mathbf{C^H}, \mathbf{C^D}\right)\right\|^2, \tag{15}$$

where $t \sim \mathrm{Uniform}\{1, \ldots, T\}, \hat{\mathbf{X}}^0 \sim q(\hat{\mathbf{X}}^0), \epsilon \sim \mathcal{N}(\mathbf{0}, \mathbf{I})$.

For data imputation, we treat all missing values as the imputation target and all the observed values as the observation condition, i.e., $\hat{\mathbf{X}} = \mathbf{X} \odot (1 - \mathbf{M})$, $\mathbf{X}^\mathbf{C} = \mathbf{X}$. We start from $\hat{\mathbf{X}}^T \sim \mathcal{N}(\mathbf{0}, \mathbf{I})$ and perform the $T$-step diffusion reverse process following Equation 14, to obtain final imputation values $\hat{\mathbf{X}}^0$. Please see the detailed training and imputation algorithms in Appendix A.3.2.

## 4 Experiment

This section experimentally evaluates both the imputation effectiveness and the improvement of real downstream applications for our FGTI, against various competing methods. All experiments are performed on a machine with Intel Core 3.0GHz i9 CPU, NVIDIA GeForce RTX 3090 24GB GPU, and 64GB RAM. The source code and datasets are available online [1].

### 4.1 Experimental Setup

#### 4.1.1 Datasets

We employ three real time series datasets with real-world missing values. **KDD** [6] collects 8,034 meteorological and air quality readings of nine stations from January 30, 2017 to January 31, 2018 in Beijing, with 4.46% real missing values. This dataset is collected every one hour and eleven sensor readings are recorded at each station. **Guangzhou** [10] records traffic speeds per ten minutes on 214 anonymous roads in Guangzhou from August 1, 2016 to September 30, 2016. There are 1.29% real missing values in the dataset. **PhysioNet** [42] contains 37 measurement readings from 11,988 patients within 48 hours of the ICU admission. 79.71% measurements are missing in the dataset, and 1,707 patients died after 48 hours of the ICU admission. Following existing studies [29; 30], since the ground truth is unavailable, we ignore these missing values when evaluating the imputation accuracy in comparative experiments and model analysis, but consider them in the application study.

Table 1: Imputation performance of various methods over real datasets with different missing rates

| Dataset | Miss. | Metric | Mean | BTMF | TIDER | BRITS | TST | SAITS | TimesNet | LaST | FreTS | GRIN | TimeCIB | GAIN | CSDI | SSSD | PriSTI | FGTI |
|---|---|---|---|---|---|---|---|---|---|---|---|---|---|---|---|---|---|---|
| KDD | 10% | RMSE | 0.993 | 0.529 | 0.777 | 0.700 | 0.594 | 0.542 | 0.484 | 0.473 | 0.630 | 0.565 | 0.589 | 0.864 | 0.459 | 0.697 | 0.472 | **0.406** |
| | | MAE | 0.718 | 0.285 | 0.527 | 0.407 | 0.360 | 0.304 | 0.313 | 0.287 | 0.412 | 0.322 | 0.367 | 0.607 | 0.177 | 0.397 | 0.169 | **0.149** |
| | 20% | RMSE | 1.007 | 0.554 | 0.797 | 0.729 | 0.740 | 0.575 | 0.542 | 0.532 | 0.741 | 0.607 | 0.613 | 0.877 | 0.500 | 0.701 | 0.534 | **0.451** |
| | | MAE | 0.718 | 0.286 | 0.531 | 0.416 | 0.371 | 0.310 | 0.307 | 0.310 | 0.489 | 0.339 | 0.369 | 0.606 | 0.187 | 0.392 | 0.180 | **0.161** |
| | 30% | RMSE | 0.997 | 0.541 | 0.783 | 0.720 | 0.642 | 0.574 | 0.578 | 0.574 | 0.796 | 0.617 | 0.603 | 0.870 | 0.519 | 0.717 | 0.547 | **0.448** |
| | | MAE | 0.717 | 0.286 | 0.528 | 0.420 | 0.376 | 0.319 | 0.357 | 0.350 | 0.546 | 0.360 | 0.370 | 0.612 | 0.199 | 0.413 | 0.195 | **0.176** |
| | 40% | RMSE | 1.001 | 0.548 | 0.790 | 0.734 | 0.702 | 0.593 | 0.648 | 0.634 | 0.850 | 0.650 | 0.611 | 0.883 | 0.569 | 0.747 | 0.581 | **0.478** |
| | | MAE | 0.718 | 0.287 | 0.532 | 0.428 | 0.387 | 0.332 | 0.418 | 0.393 | 0.591 | 0.387 | 0.372 | 0.623 | 0.220 | 0.435 | 0.217 | **0.205** |
| Guang. | 10% | RMSE | 0.799 | 0.384 | 0.549 | 0.481 | 0.368 | 0.417 | 0.400 | 0.347 | 0.456 | 0.466 | 0.451 | 0.804 | 0.306 | 0.434 | 0.242 | **0.230** |
| | | MAE | 0.592 | 0.252 | 0.392 | 0.299 | 0.249 | 0.264 | 0.270 | 0.244 | 0.340 | 0.354 | 0.300 | 0.550 | 0.210 | 0.293 | **0.170** | **0.170** |
| | 20% | RMSE | 0.799 | 0.384 | 0.537 | 0.481 | 0.398 | 0.415 | 0.433 | 0.440 | 0.602 | 0.501 | 0.448 | 0.804 | 0.324 | 0.460 | 0.324 | **0.258** |
| | | MAE | 0.592 | 0.252 | 0.382 | 0.300 | 0.275 | 0.264 | 0.303 | 0.312 | 0.460 | 0.385 | 0.298 | 0.550 | 0.220 | 0.315 | 0.197 | **0.176** |
| | 30% | RMSE | 0.799 | 0.384 | 0.536 | 0.485 | 0.442 | 0.420 | 0.481 | 0.545 | 0.709 | 0.542 | 0.448 | 0.805 | 0.364 | 0.545 | 0.510 | **0.291** |
| | | MAE | 0.592 | 0.252 | 0.382 | 0.301 | 0.312 | 0.267 | 0.348 | 0.388 | 0.547 | 0.419 | 0.298 | 0.551 | 0.242 | 0.384 | 0.271 | **0.202** |
| | 40% | RMSE | 0.800 | 0.385 | 0.541 | 0.491 | 0.540 | 0.422 | 0.542 | 0.637 | 0.787 | 0.584 | 0.449 | 0.807 | 0.439 | 0.622 | 0.650 | **0.356** |
| | | MAE | 0.592 | 0.253 | 0.387 | 0.306 | 0.397 | 0.270 | 0.401 | 0.458 | 0.611 | 0.455 | 0.299 | 0.554 | 0.283 | 0.444 | 0.381 | **0.254** |
| Phy. | 10% | RMSE | 0.932 | 0.630 | 0.879 | 0.732 | 0.632 | 0.645 | 0.776 | 0.768 | 0.804 | 0.682 | 0.697 | 1.006 | 0.619 | 0.875 | 0.652 | **0.580** |
| | | MAE | 0.678 | 0.348 | 0.605 | 0.446 | 0.389 | 0.371 | 0.525 | 0.516 | 0.540 | 0.424 | 0.450 | 0.747 | 0.310 | 0.528 | 0.369 | **0.286** |
| | 20% | RMSE | 0.935 | 0.627 | 0.889 | 0.718 | 0.640 | 0.641 | 0.806 | 0.786 | 0.825 | 0.670 | 0.683 | 0.988 | 0.664 | 0.834 | 0.638 | **0.577** |
| | | MAE | 0.675 | 0.362 | 0.624 | 0.451 | 0.417 | 0.384 | 0.569 | 0.550 | 0.576 | 0.434 | 0.455 | 0.740 | 0.335 | 0.507 | 0.376 | **0.309** |
| | 30% | RMSE | 0.934 | 0.658 | 0.911 | 0.734 | 0.688 | 0.670 | 0.849 | 0.825 | 0.861 | 0.695 | 0.697 | 0.995 | 0.805 | 0.882 | 0.661 | **0.624** |
| | | MAE | 0.676 | 0.382 | 0.638 | 0.457 | 0.452 | 0.404 | 0.600 | 0.578 | 0.603 | 0.446 | 0.459 | 0.738 | 0.360 | 0.545 | 0.387 | **0.336** |
| | 40% | RMSE | 0.932 | 0.677 | 0.935 | 0.739 | 0.732 | 0.688 | 0.872 | 0.850 | 0.883 | 0.708 | 0.698 | 0.983 | 0.705 | 0.904 | 0.679 | **0.669** |
| | | MAE | 0.677 | 0.412 | 0.658 | 0.466 | 0.493 | 0.431 | 0.623 | 0.603 | 0.626 | 0.464 | 0.466 | 0.729 | 0.395 | 0.555 | 0.406 | **0.376** |

### 4.1.2 Criteria

Following previous studies [33; 27], we employ RMSE [22] and MAE [7] to evaluate the imputation accuracy. For both, the smaller the value is, the more effective the imputation will be. For the air quality prediction application over the KDD dataset in Section 4.5, we also use RMSE as the metric. In addition, the AUC score [32] is used to measure the patient mortality forecasting application over the PhysioNet dataset. The large the value is, the better the forecasting result will be.

### 4.1.3 Baselines

We compare with fifteen widely adopted time series imputation methods, including statistics-based Mean [41], matrix factorization based BTMF [9] and TIDER [28], deep learning based BRITS [6], TST [57], SAITS [14], TimesNet [52], LaST [50] and FreTS [55], GNN-based GRIN [12], VAE-based TimeCIB [11]. GAN-based GAIN [56], Diffusion-based CSDI [44], SSSD [2], and PriSTI [27]. For methods such as GRIN and PriSTI that require an adjacency matrix as input to show relationships between attributes, we use the identity matrix by default, where every attribute has dependencies with others in the time series. Since LaST and FreTS only focus on the time series forecasting task, we adapt them to the imputation task based on the seting of TimesNet. For methods in which the authors recommend parameters such as SAITS, MIWAE, GPVAE, CSDI and PRiSTI, we use these parameters as suggested. The other methods are also configured in a best-effort fashion by iteratively choosing good parameters.

### 4.2 Comparative Experiments

We first explore the imputation performance of different methods over real datasets with different missing rates. The observed values with various missing rates are randomly removed under the missing completely at random (MCAR) mechanism [5] to form the imputation target. Each experiment is repeated five times with different generated missing values and random seeds, and the average result is reported in Table 1. Note that RMSE reflects the absolute difference between the imputation value

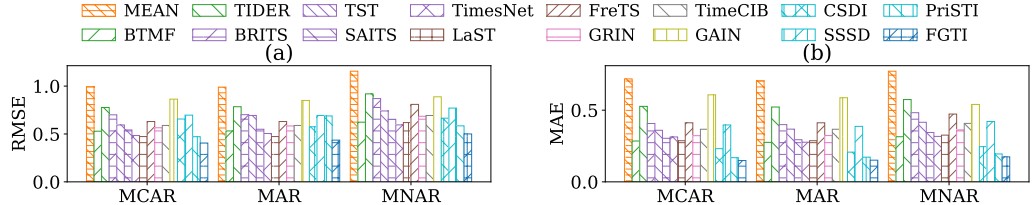

Figure 4: Varying the missing mechanism over KDD dataset with 10% missing values

Table 2: Ablation analysis of FGTI with 10% missing values

| Method | KDD | | Guangzhou | | PhysioNet | |
|---|---|---|---|---|---|---|
| | RMSE | MAE | RMSE | MAE | RMSE | MAE |
| w/o Cross-domain | 0.4210 | 0.1603 | 0.2365 | 0.1607 | 0.6288 | 0.3748 |
| w/o Frequency condition | 0.4151 | 0.1505 | 0.2402 | 0.1635 | 0.6165 | 0.3719 |
| w/o Dominant-frequency filter | 0.4151 | 0.1518 | 0.2383 | 0.1625 | 0.6540 | 0.3799 |
| w/o High-frequency filter | 0.4128 | 0.1493 | 0.2367 | 0.1611 | 0.7294 | 0.3654 |
| FGTI | **0.4057** | **0.1489** | **0.2325** | **0.1584** | **0.5801** | **0.2856** |

and ground truth in the context of the data scale, and is not bounded by a specific range. The missing rates in the table are with respect to the observed values.

We can find that our method achieves the best imputation accuracy under various missing rates. When there is more missing data, deep learning based imputation models are less accurate due to the lack of observation condition information. Nevertheless, our approach uses frequency domain information and achieves superior results. Our FGTI model surpasses the state-of-the-art generative imputation models in various cases, thanks to the incorporation of high-frequency and dominant-frequency condition information.

Moreover, since missing data are usually associated with the environment in reality, we consider two additional typical missing data injection mechanisms following the same line of the existing study [33], i.e., missing at random (MAR) [53] and missing not at random (MNAR) [45]. Specifically, the probability of missing data in MAR is higher when the temperature reading is low in the KDD dataset. On the other hand, in the MNAR scenario, there is a higher probability of missing data during the periods when the reading of each feature is lower. Figure 4 and Figures 8-9 in Appendix A.4.1 show the corresponding imputation results. One can find that the imputation results under different missing mechanisms are relatively similar, and our FGTI achieves optimal performance consistently. This result demonstrates that FGTI can handle missing data in various missing scenarios.

### 4.3 Ablation Analysis

We explore the effect of different elements in our FGTI on imputation performance through the following four ablation scenarios. **(1) w/o Cross-domain**: No extra condition representation is provided for cross-domain representation learning frameworks, where the fusion processes degrade to the standard self-attention. This scenario is used to validate the role of cross-domain representation for imputation. **(2) w/o Frequency condition**: In this scenario, the frequency-domain information is absent, and only the observations in the time-domain are utilized as the condition, to investigate the impact of the frequency-domain information. **(3) w/o Dominant-frequency filter**: Remove the dominant-frequency filter from the structure to observe its impact on data imputation performance. **(4) w/o High-frequency filter**: This case exemplifies how crucial the high-frequency filter is, by removing it from the pipeline.

Based on the results presented in Table 2, it is evident that cross-domain representation learning frameworks and the two types of frequency-domain information are crucial in the process of imputation, which verifies the necessity of each component in our FGTI.

### 4.4 Resource Consumption

We present the resource consumption results of different methods in Figure 5. It can be observed that the running time of FGTI is roughly at the same level as other diffusion-based methods. The overall

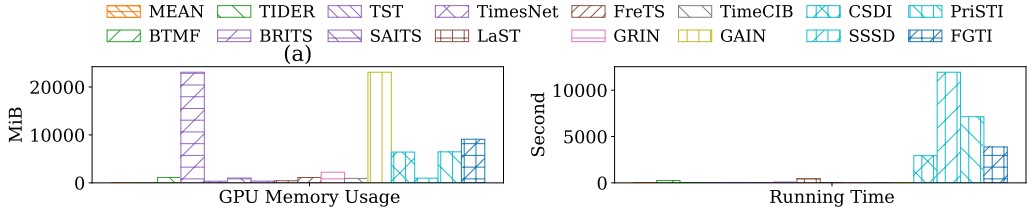

Figure 5: Resource consumption over KDD dataset with 10% missing values

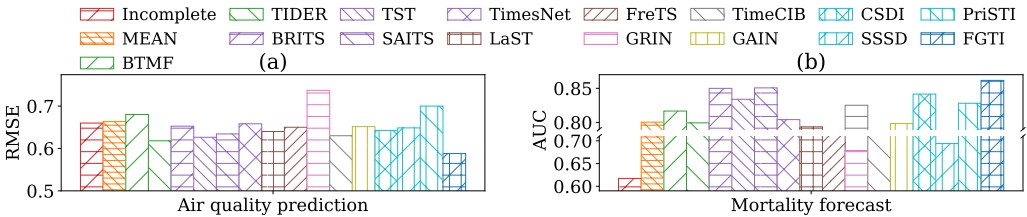

Figure 6: Application results of air quality prediction over KDD dataset and mortality forecast over Physionet dataset

resource consumption of FGTI, implemented based on the diffusion model, is slightly higher than that of CSDI, due to the inclusion of high-frequency information and dominant-frequency information. However, since FGTI can achieve better imputation results than other methods, as shown in Table 1, we argue that it is acceptable to incur such an extra resource consumption.

## 4.5 Application Study

To validate the effectiveness of applying imputation in real-world downstream applications, we consider air quality prediction and mortality forecasting tasks. For the air quality prediction application, we first impute real-world missing data in the KDD dataset by various imputation methods. Then, we analyze the records in the previous twelve hours and use the AdaBoost regressor [37] to estimate the average PM2.5 concentration for the upcoming six hours. Figure 6(a) shows that our method achieves the highest improvement in the air quality prediction task. Figure 6(b) reports the mortality forecast performance over the PhysioNet dataset. We train the MLP classifier [37] to forecast the mortality on the data without/with imputation. As shown, FGTI achieves the best performance again, which verifies the applicability of our work. Notably, various imputation methods provide a noteworthy and favorable impact on the forecast task, which demonstrates the necessity of imputation.

## 5 Conclusion

In this paper, we study imputing incomplete multivariate time series data, through reducing the imputation error in the residual term. By effectively incorporating frequency-domain insights into the generative framework, our FGTI surpasses existing models by capturing high-frequency information and dominant-frequency information. The introduced cross-domain representation learning frameworks further enhance its capability to handle time and attribute dependencies. Comprehensive experimental evaluations over real-world incomplete datasets demonstrate the superiority of FGTI in both the imputation accuracy and the improvement of downstream applications.

**Acknowledgements**

This work is supported in part by the Fundamental Research Funds for the Central Universities, Nankai University (63231147), the National Natural Science Foundation of China (62302241, 62306085, 62372252, 72342017), Shenzhen College Stability Support Plan (GXWD20231130151329002).

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

# A  Appendix

## A.1  Proof of Proposition 3.1

We use the conditional entropy in information theory to reflect the amount of uncertainty. For the reverse process in [20], the imputation target $\hat{\mathbf{X}}^t$ is specified as the condition, thus we use

$$\mathrm{H}\left(\hat{\mathbf{X}}^{t-1} \mid \hat{\mathbf{X}}^{t}\right) = -\int p_\theta\left(\hat{\mathbf{X}}^{t-1}, \hat{\mathbf{X}}^{t}\right) \log p_\theta\left(\hat{\mathbf{X}}^{t-1} \mid \hat{\mathbf{X}}^{t}\right) \mathrm{d}\hat{\mathbf{X}}^{t-1}$$

to model the uncertainty of the reverse process of DDPM.

Similarly, we use $\mathrm{H}\left(\hat{\mathbf{X}}^{t-1} \mid \hat{\mathbf{X}}^{t}, \mathbf{X}^{\mathbf{C}}\right)$ to model the uncertainty of the reverse process of the CSDI [44] model, which only uses the observations as the condition. For our FGTI, we utilize $\mathrm{H}\left(\hat{\mathbf{X}}^{t-1} \mid \hat{\mathbf{X}}^{t}, \mathbf{X}^{\mathbf{C}}, \mathbf{C}^{\mathbf{H}}, \mathbf{C}^{\mathbf{D}}\right)$.

According to the property of the conditional entropy, we first have

$$\mathrm{H}\left(\hat{\mathbf{X}}^{t-1} \mid \hat{\mathbf{X}}^{t}\right) \leq \mathrm{H}\left(\hat{\mathbf{X}}^{t-1}\right).$$

Using the definition of mutual information, we have

$$\mathrm{I}\left(\hat{\mathbf{X}}^{t-1}; \hat{\mathbf{X}}^{t}\right) = \mathrm{H}\left(\hat{\mathbf{X}}^{t-1}\right) - \mathrm{H}\left(\hat{\mathbf{X}}^{t-1} \mid \hat{\mathbf{X}}^{t}\right).$$

From Equation 12, we know that $\mathrm{I}\left(\hat{\mathbf{X}}^{t-1}; \hat{\mathbf{X}}^{t}\right) > 0$, we thus have

$$\mathrm{H}\left(\hat{\mathbf{X}}^{t-1} \mid \hat{\mathbf{X}}^{t}\right) < \mathrm{H}\left(\hat{\mathbf{X}}^{t-1}\right).$$

According to the chain rule of the entropy, we have

$$\mathrm{H}\left(\hat{\mathbf{X}}^{t-1}, \hat{\mathbf{X}}^{t}, \mathbf{X}^{\mathbf{C}}\right) = \mathrm{H}\left(\hat{\mathbf{X}}^{t-1} \mid \hat{\mathbf{X}}^{t}, \mathbf{X}^{\mathbf{C}}\right) + \mathrm{H}\left(\hat{\mathbf{X}}^{t}, \mathbf{X}^{\mathbf{C}}\right).$$

Then we can get

$$\mathrm{H}\left(\hat{\mathbf{X}}^{t-1} \mid \hat{\mathbf{X}}^{t}, \mathbf{X}^{\mathbf{C}}\right) = \mathrm{H}\left(\mathbf{X}^{\mathbf{C}} \mid \hat{\mathbf{X}}^{t-1}, \hat{\mathbf{X}}^{t}\right) + \mathrm{H}\left(\hat{\mathbf{X}}^{t-1}, \hat{\mathbf{X}}^{t}\right) - \mathrm{H}\left(\mathbf{X}^{\mathbf{C}} \mid \hat{\mathbf{X}}^{t}\right) - \mathrm{H}\left(\hat{\mathbf{X}}^{t}\right)$$
$$= \mathrm{H}\left(\hat{\mathbf{X}}^{t-1} \mid \hat{\mathbf{X}}^{t}\right) + \mathrm{H}\left(\mathbf{X}^{\mathbf{C}} \mid \hat{\mathbf{X}}^{t-1}, \hat{\mathbf{X}}^{t}\right) - \mathrm{H}\left(\mathbf{X}^{\mathbf{C}} \mid \hat{\mathbf{X}}^{t}\right).$$

According to the Equation 12, $\hat{\mathbf{X}}^{t-1}$ adds the noise one less time than $\hat{\mathbf{X}}^{t}$, which indicating that $\hat{\mathbf{X}}^{t-1}$ is closer to the observations. Thus, we have

$$\mathrm{H}\left(\mathbf{X}^{\mathbf{C}} \mid \hat{\mathbf{X}}^{t-1}, \hat{\mathbf{X}}^{t}\right) < \mathrm{H}\left(\mathbf{X}^{\mathbf{C}} \mid \hat{\mathbf{X}}^{t}\right).$$

By substituting this, we can obtain

$$\mathrm{H}\left(\hat{\mathbf{X}}^{t-1} \mid \hat{\mathbf{X}}^{t}, \mathbf{X}^{\mathbf{C}}\right) < \mathrm{H}\left(\hat{\mathbf{X}}^{t-1} \mid \hat{\mathbf{X}}^{t}\right).$$

Following the same line, we can also derive that

$$\mathrm{H}\left(\hat{\mathbf{X}}^{t-1} \mid \hat{\mathbf{X}}^{t}, \mathbf{X}^{\mathbf{C}}, \mathbf{C}^{\mathbf{H}}, \mathbf{C}^{\mathbf{D}}\right) < \mathrm{H}\left(\hat{\mathbf{X}}^{t-1} \mid \hat{\mathbf{X}}^{t}, \mathbf{X}^{\mathbf{C}}\right).$$

This result implies that adding $\mathbf{C}^{\mathbf{H}}$ and $\mathbf{C}^{\mathbf{D}}$ to the condition can simplify the distribution that the diffusion models need to learn by reducing the entropy. This simplification can lead to more efficient learning and improve the model's imputation performance by narrowing down the scope of the target distribution's randomness and making its outcomes more predictable.

## A.2 Mathematical details of FGTI

### A.2.1 Details of Equation 13

If the complete imputation target distribution $q(\hat{\mathbf{X}}^0)$ is known, we can first get a sampled complete imputation target $\hat{\mathbf{X}}^0 \sim q(\hat{\mathbf{X}}^0)$. Following Equation 12, we can obtain $\hat{\mathbf{X}}^t$ by

$$\hat{\mathbf{X}}^t = \sqrt{\alpha^t}\hat{\mathbf{X}}^{t-1} + \sqrt{1-\alpha^t}\epsilon^t, \epsilon^t \sim \mathcal{N}(\mathbf{0}, \mathbf{I}).$$

Similarly, we can also obtain $\hat{\mathbf{X}}^{t-1}$ by

$$\hat{\mathbf{X}}^{t-1} = \sqrt{\alpha^{t-2}}\hat{\mathbf{X}}^{t-2} + \sqrt{1-\alpha^{t-1}}\epsilon^{t-1}, \epsilon^{t-1} \sim \mathcal{N}(\mathbf{0}, \mathbf{I}).$$

Combining the above two equations, we get

$$\hat{\mathbf{X}}^t = \sqrt{\alpha^t}\left(\sqrt{\alpha^{t-1}}\hat{\mathbf{X}}^{t-2} + \sqrt{1-\alpha^{t-1}}\epsilon^{t-1}\right) + \sqrt{1-\alpha^t}\epsilon^t$$
$$= \sqrt{\alpha^t\alpha^{t-1}}\hat{\mathbf{X}}^{t-2} + \left(\sqrt{\alpha^t(1-\alpha^{t-1})}\epsilon^{t-1} + \sqrt{1-\alpha^t}\epsilon^t\right).$$

As $\epsilon^{t-1} \sim \mathcal{N}(\mathbf{0}, \mathbf{I})$, $\epsilon^t \sim \mathcal{N}(\mathbf{0}, \mathbf{I})$, we can infer that $\sqrt{\alpha^t(1-\alpha^{t-1})}\epsilon^{t-1} \sim \mathcal{N}(\mathbf{0}, [\alpha^t(1-\alpha^{t-1})]\mathbf{I})$, $\sqrt{1-\alpha^t}\epsilon^t \sim \mathcal{N}(\mathbf{0}, (1-\alpha^t)\mathbf{I})$. Therefore, we can get

$$\hat{\mathbf{X}}^t = \sqrt{\alpha^t\alpha^{t-1}}\hat{\mathbf{X}}^{t-2} + \sqrt{1-\alpha^t\alpha^{t-1}}\epsilon$$
$$= \sqrt{\alpha^t\alpha^{t-1}\alpha^{t-2}}\hat{\mathbf{X}}^{t-3} + \sqrt{1-\alpha^t\alpha^{t-1}\alpha^{t-1}}\epsilon$$
$$= \dots$$
$$= \sqrt{\prod_{i=1}^t \alpha^i}\hat{\mathbf{X}}^0 + \sqrt{1-\prod_{i=1}^t \alpha^i}\epsilon,$$

where $\epsilon \sim \mathcal{N}(\mathbf{0}, \mathbf{I})$.

### A.2.2 Details of the Parameterization

Combining the Bayes' theorem, we start by

$$p_\theta(\hat{\mathbf{X}}^{t-1} \mid \hat{\mathbf{X}}^t, \mathbf{X}^{\mathbf{C}}, \mathbf{C}^{\mathbf{H}}, \mathbf{C}^{\mathbf{D}}) = p_\theta(\hat{\mathbf{X}}^t \mid \hat{\mathbf{X}}^{t-1}, \mathbf{X}^{\mathbf{C}}, \mathbf{C}^{\mathbf{H}}, \mathbf{C}^{\mathbf{D}})\frac{p_\theta(\hat{\mathbf{X}}^{t-1} \mid \mathbf{X}^{\mathbf{C}}, \mathbf{C}^{\mathbf{H}}, \mathbf{C}^{\mathbf{D}})}{p_\theta(\hat{\mathbf{X}}^t \mid \mathbf{X}^{\mathbf{C}}, \mathbf{C}^{\mathbf{H}}, \mathbf{C}^{\mathbf{D}})}.$$

According to Equation 12, the expected $p_\theta(\hat{\mathbf{X}}^t \mid \hat{\mathbf{X}}^{t-1}, \mathbf{X}^{\mathbf{C}}, \mathbf{C}^{\mathbf{H}}, \mathbf{C}^{\mathbf{D}})$ is

$$p_\theta(\hat{\mathbf{X}}^t \mid \hat{\mathbf{X}}^{t-1}, \mathbf{X}^{\mathbf{C}}, \mathbf{C}^{\mathbf{H}}, \mathbf{C}^{\mathbf{D}}) \sim \mathcal{N}(\sqrt{\alpha^t}\hat{\mathbf{X}}^{t-1}, (1-\alpha^t)\mathbf{I}).$$

Furthermore, by incorporating Equation 13, we can obtain

$$p_\theta(\hat{\mathbf{X}}^{t-1} \mid \mathbf{X}^{\mathbf{C}}, \mathbf{C}^{\mathbf{H}}, \mathbf{C}^{\mathbf{D}}) \sim \mathcal{N}(\sqrt{\overline{\alpha^{t-1}}}\hat{\mathbf{X}}^0_\theta, (1-\sqrt{\overline{\alpha^{t-1}}})\mathbf{I}),$$

$$p_\theta(\hat{\mathbf{X}}^t \mid \mathbf{X}^{\mathbf{C}}, \mathbf{C}^{\mathbf{H}}, \mathbf{C}^{\mathbf{D}}) \sim \mathcal{N}(\sqrt{\overline{\alpha^t}}\hat{\mathbf{X}}^0_\theta, (1-\sqrt{\overline{\alpha^t}})\mathbf{I}),$$

where $\hat{\mathbf{X}}^0_\theta$ is a virtual result that is expected to be obtained through Equation 14. Combining Equation 12, we can parameterize it by $\hat{\mathbf{X}}^t = \sqrt{\overline{\alpha^t}}\hat{\mathbf{X}}^0_\theta + \sqrt{1-\overline{\alpha^t}}\epsilon_\theta\left(t, \hat{\mathbf{X}}^t, \mathbf{X}^{\mathbf{C}}, \mathbf{C}^{\mathbf{H}}, \mathbf{C}^{\mathbf{D}}\right)$, where $\epsilon_\theta(\cdot)$ outputs the predicted added noise to $\hat{\mathbf{X}}^{t-1}$ based on the parameter set $\theta$. In other words, $\hat{\mathbf{X}}^0_\theta = \frac{1}{\sqrt{\overline{\alpha^t}}}\left[\hat{\mathbf{X}}^t - \sqrt{1-\overline{\alpha^t}}\epsilon_\theta\left(t, \hat{\mathbf{X}}^t, \mathbf{X}^{\mathbf{C}}, \mathbf{C}^{\mathbf{H}}, \mathbf{C}^{\mathbf{D}}\right)\right].$

By merging the above three distributions, we can derive

$$p_\theta(\hat{\mathbf{X}}^{t-1} \mid \hat{\mathbf{X}}^t, \mathbf{X}^{\mathbf{C}}, \mathbf{C}^{\mathbf{H}}, \mathbf{C}^{\mathbf{D}}) \propto \exp\left\{ -\frac{1}{2}\left[ \frac{(\hat{\mathbf{X}}^t - \sqrt{\alpha^t}\hat{\mathbf{X}}^{t-1})^2}{1-\alpha^t} + \frac{(\hat{\mathbf{X}}^{t-1} - \sqrt{\overline{\alpha^{t-1}}}\hat{\mathbf{X}}_\theta^0)^2}{1-\overline{\alpha^{t-1}}} \right.\right.$$

$$\left.\left. - \frac{(\hat{\mathbf{X}}^t - \sqrt{\overline{\alpha^t}}\hat{\mathbf{X}}_\theta^0)^2}{1-\overline{\alpha^t}} \right]\right\}$$

$$= \exp\left\{ -\frac{1}{2}\left[ \left(\frac{\alpha^t}{\beta^t} + \frac{1}{1-\overline{\alpha^{t-1}}}\right)(\hat{\mathbf{X}}^{t-1})^2 \right.\right.$$

$$\left.\left. - \left(\frac{2\sqrt{\alpha^t}\hat{\mathbf{X}}^t}{\beta^t} + \frac{2\sqrt{\overline{\alpha^{t-1}}}\hat{\mathbf{X}}_\theta^0}{1-\overline{\alpha^{t-1}}}\right)\hat{\mathbf{X}}^{t-1} + \cdots \right]\right\},$$

where $(\hat{\mathbf{X}}^{t-1})^2$ denotes the inner product of $\hat{\mathbf{X}}^{t-1}$.

On the other hand, from the probability density function of the Gaussian distribution, we can also obtain

$$p_\theta(\hat{\mathbf{X}}^{t-1} \mid \hat{\mathbf{X}}^t, \mathbf{X}^{\mathbf{C}}, \mathbf{C}^{\mathbf{H}}, \mathbf{C}^{\mathbf{D}}) \propto \exp\left\{ -\frac{1}{2}\frac{\left(\hat{\mathbf{X}}^{t-1} - \mu_\theta\left[\hat{\mathbf{X}}^{t-1} \mid \hat{\mathbf{X}}^t, \mathbf{X}^{\mathbf{C}}, \mathbf{C}^{\mathbf{H}}, \mathbf{C}^{\mathbf{D}}\right]\right)^2}{[\sigma^{t-1}]^2} \right\}$$

$$= \exp\left\{ -\frac{1}{2}\left[ \frac{1}{[\sigma^{t-1}]^2}(\hat{\mathbf{X}}^{t-1})^2 \right.\right.$$

$$\left.\left. + \frac{2\mu_\theta\left[\hat{\mathbf{X}}^{t-1} \mid \hat{\mathbf{X}}^t, \mathbf{X}^{\mathbf{C}}, \mathbf{C}^{\mathbf{H}}, \mathbf{C}^{\mathbf{D}}\right]}{[\sigma^{t-1}]^2}\hat{\mathbf{X}}^{t-1} + \cdots \right]\right\}.$$

Therefore, we can get

$$\frac{1}{[\sigma^{t-1}]^2} = \frac{\alpha^t}{\beta^t} + \frac{1}{1-\overline{\alpha^{t-1}}},$$

$$\frac{2\mu_\theta\left[\hat{\mathbf{X}}^{t-1} \mid \hat{\mathbf{X}}^t, \mathbf{X}^{\mathbf{C}}, \mathbf{C}^{\mathbf{H}}, \mathbf{C}^{\mathbf{D}}\right]}{[\sigma^{t-1}]^2} = \frac{2\sqrt{\alpha^t}\hat{\mathbf{X}}^t}{\beta^t} + \frac{2\sqrt{\overline{\alpha^{t-1}}}\hat{\mathbf{X}}_\theta^0}{1-\overline{\alpha^{t-1}}}.$$

We can first obtain the following result from the first equation as $\alpha^t = 1 - \beta^t$

$$[\sigma^{t-1}]^2 = \frac{(1-\overline{\alpha^{t-1}})\beta^t}{1-\overline{\alpha^t}}.$$

After that, we take it into the second equation

$$\frac{\mu_\theta\left[\hat{\mathbf{X}}^{t-1} \mid \hat{\mathbf{X}}^t, \mathbf{X}^{\mathbf{C}}, \mathbf{C}^{\mathbf{H}}, \mathbf{C}^{\mathbf{D}}\right](1-\overline{\alpha^t})}{(1-\overline{\alpha^{t-1}})\beta^t} = \frac{\sqrt{\alpha^t}\hat{\mathbf{X}}^t(1-\overline{\alpha^t}) + \sqrt{\overline{\alpha^{t-1}}}\hat{\mathbf{X}}_\theta^0\beta^t}{(1-\overline{\alpha^{t-1}})\beta^t}.$$

As the virtual result $\hat{\mathbf{X}}_\theta^0 = \frac{1}{\sqrt{\overline{\alpha^t}}}\left[\hat{\mathbf{X}}^t - \sqrt{1-\overline{\alpha^t}}\epsilon_\theta\left(t, \hat{\mathbf{X}}^t, \mathbf{X}^{\mathbf{C}}, \mathbf{C}^{\mathbf{H}}, \mathbf{C}^{\mathbf{D}}\right)\right]$, we have

$$\mu_\theta\left[\hat{\mathbf{X}}^{t-1} \mid \hat{\mathbf{X}}^t, \mathbf{X}^{\mathbf{C}}, \mathbf{C}^{\mathbf{H}}, \mathbf{C}^{\mathbf{D}}\right] = \frac{\sqrt{\alpha^t}(1-\overline{\alpha^{t-1}})}{1-\overline{\alpha^t}}\hat{\mathbf{X}}^t + \frac{\sqrt{\overline{\alpha^{t-1}}}\beta^t}{1-\overline{\alpha^t}}\frac{1}{\sqrt{\overline{\alpha^{t-1}}}}\hat{\mathbf{X}}^t$$

$$- \frac{\sqrt{\overline{\alpha^{t-1}}}\beta^t}{1-\overline{\alpha^t}}\frac{\sqrt{1-\overline{\alpha^t}}}{\sqrt{\overline{\alpha^{t-1}}}}\epsilon_\theta\left(t, \hat{\mathbf{X}}^t, \mathbf{X}^{\mathbf{C}}, \mathbf{C}^{\mathbf{H}}, \mathbf{C}^{\mathbf{D}}\right)$$

$$= \frac{1}{\sqrt{\overline{\alpha^t}}}\left[\hat{\mathbf{X}}^t - \frac{\beta^t}{\sqrt{1-\overline{\alpha^t}}}\epsilon_\theta\left(t, \hat{\mathbf{X}}^t, \mathbf{X}^{\mathbf{C}}, \mathbf{C}^{\mathbf{H}}, \mathbf{C}^{\mathbf{D}}\right)\right].$$

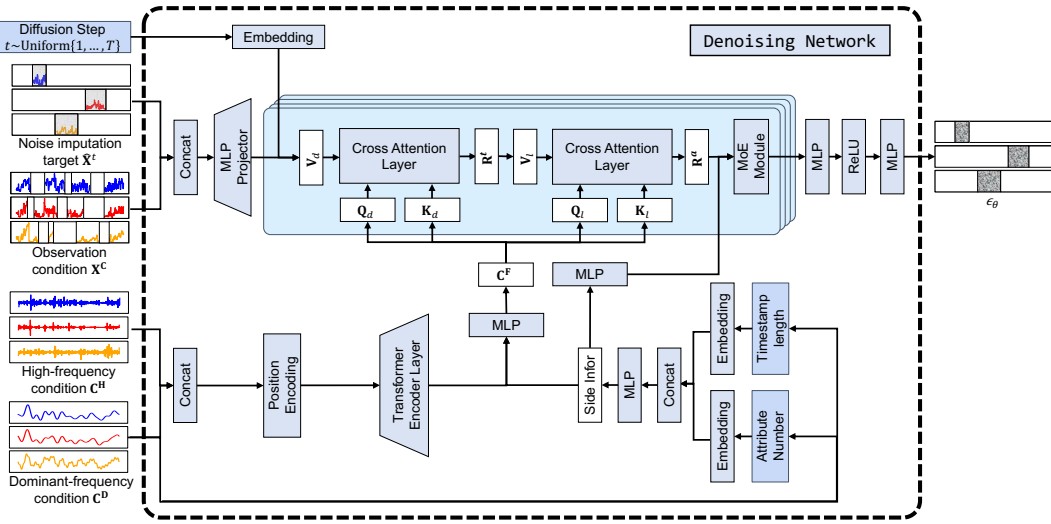

Figure 7: Architecture of the Denoising Network $\epsilon_\theta(\cdot)$ in FGTI

## A.3 Implementation Details

### A.3.1 Detailed Architecture of the Denoising Network

In this section, we present the detailed architecture of the denoising network $\epsilon_\theta(\cdot)$ in FGTI model. As shown in Figure 7, the input projector of the noise imputation target and the observation condition is an MLP layer, and the output projector is a 2-layer MLP with ReLU activation function. For the encoder of the frequency-domain information, it is implemented by a transformer backbone consisting of the position encoding layer and the transformer encoder layer.

### A.3.2 Algorithms

---

**Algorithm 1** Training process of FGTI implemented by the diffusion model

---

**Input:** Incomplete time series $\mathbf{X}$, the number of diffusion step $T$
**Output:** Optimized denoising network $\epsilon_\theta(\cdot)$

1: **repeat**
2:     $\hat{\mathbf{X}}^0 \leftarrow$ select observed values in $\mathbf{X}$
3:     $t \sim \text{Uniform}\{1, \ldots, T\}$
4:     $\epsilon \sim \mathcal{N}(\mathbf{0}, \mathbf{I})$
5:     $\hat{\mathbf{X}}^t \leftarrow \sqrt{\overline{\alpha^t}}\hat{\mathbf{X}}^0 + \sqrt{1 - \overline{\alpha^t}}\epsilon$
6:     Perform Gradient Descent by $\nabla\mathcal{L}_\theta = \nabla_\theta \left\| \epsilon - \epsilon_\theta\left(t, \hat{\mathbf{X}}^t, \mathbf{X}^{\mathbf{C}}, \mathbf{C}^{\mathbf{H}}, \mathbf{C}^{\mathbf{D}}\right) \right\|^2$
7: **until** converged

---

In this section, we provide the detailed training process of our proposed FGTI model implemented by the diffusion model in Algorithm 1, and the imputation process in Algorithm 2.

## A.4 Supplemental Experiments

### A.4.1 Missing Mechanisms

In this section, we explore the imputation performance of the missing at random (MAR) [53] and missing not at random (MNAR) [45] missing mechanisms over the Guangzhou dataset and the PhysioNet dataset.

---

**Algorithm 2** Imputation process of FGTI implemented by the diffusion model

---

**Input:** A incomplete time series sample $\mathbf{X}$, the number of diffusion step $T$, the optimized denoising network $\epsilon_\theta(\cdot)$

**Output:** Filled missing values $\hat{\mathbf{X}}^0$

---

1: $\hat{\mathbf{X}} \leftarrow$ missing values in $\mathbf{X}$
2: $\hat{\mathbf{X}}^T \sim \mathcal{N}(\mathbf{0}, \mathbf{I})$
3: **for** $t = T, \ldots, 1$ **do**
4:     **if** $t > 1$ **then**
5:         $\epsilon \sim \mathcal{N}(\mathbf{0}, \mathbf{I})$
6:     **else**
7:         $\epsilon \leftarrow \mathbf{0}$
8:     **end if**
9:     $\hat{\mathbf{X}}^{t-1} \leftarrow \frac{1}{\sqrt{\alpha^t}} \left[ \hat{\mathbf{X}}^t - \frac{\beta^t}{\sqrt{1-\overline{\alpha^t}}} \epsilon_\theta \left( t, \hat{\mathbf{X}}^t, \mathbf{X}, \mathbf{C^H}, \mathbf{C^D} \right) \right] + \sqrt{\frac{(1-\overline{\alpha^{t-1}})\beta^t}{1-\overline{\alpha^t}}} \epsilon$
10: **end for**

---

As shown in Figure 8 and Figure 9, FGTI consistently achieves optimal performance, demonstrating its ability to handle missing data in various scenarios.

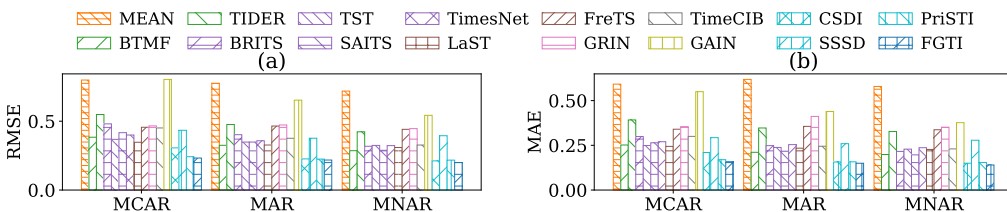

Figure 8: Varying the missing mechanism over Guangzhou dataset with 10% missing values

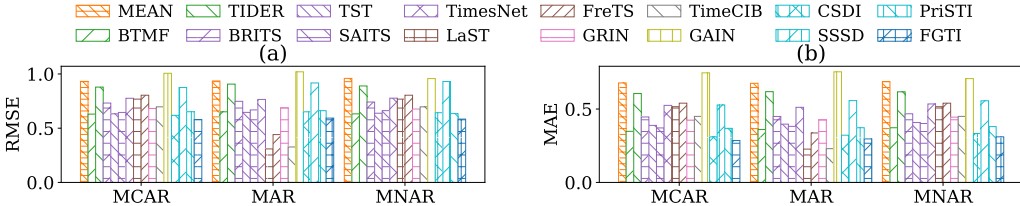

Figure 9: Varying the missing mechanism over PhysioNet dataset with 10% missing values

### A.4.2  Hyperparameter Evaluation

In this section, we perform parameter sensitivity experiments on two critical hyperparameters: the cutoff frequency of the high-frequency filter and the maximum magnitude frequency number of the dominant-frequency filter.

**Effect of the cutoff frequency $\mathcal{F}$.**  Figure 10 shows the imputation results with various cutoff frequencies $\mathcal{F}$ of the high-frequency filter. It can be found that if $\mathcal{F}$ is too small, the model may not be able to accurately capture the high-frequency information necessary for guiding the imputation of the time series residual term. The reason is that the high-frequency filter output may include too much low-frequency information. Conversely, if $\mathcal{F}$ is too large, the model cannot obtain enough high-frequency information to guide the imputation of the residual term.

**Effect of the maximum magnitude frequency number $\kappa$.**  We investigate the imputation results when adjusting the maximum amplitude frequency number $\kappa$ used for the dominant-frequency filter in Figure 11. As shown in the figure, the imputation model cannot perform well with a small $\kappa$. This is because the dominant-frequency filter cannot obtain enough smoothing information, which causes

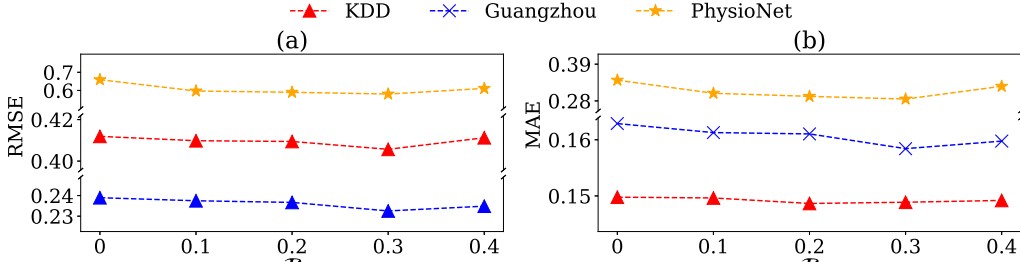

Figure 10: Varying the cutoff frequency $\mathcal{F}$ of the high-frequency filter with 10% missing values

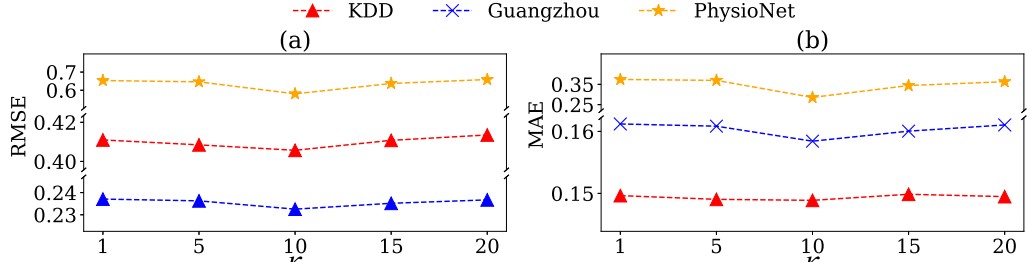

Figure 11: Varying the number of maximum magnitude frequency $\kappa$ of the dominant-frequency filter with 10% missing values

high-frequency signals to interfere with the imputation of the trend and seasonal terms of the time series. On the contrary, if $\kappa$ is too large, it can cause some high-frequency information to mix with the output condition of the dominant-frequency filter. This can prevent the model from effectively obtaining background structure information needed for imputing the trend and seasonal terms. As a result, the imputation result for extreme cases may not be optimal.

Based on the experimental results, we set the cutoff frequency $\mathcal{F}$ of the high-frequency filter to 0.3 and set the number of maximum magnitude frequency $\kappa$ of the dominant-frequency filter to 10. In addition, for other settings related to the diffusion model, we adopt hyperparameters recommended by the existing well-established models [44; 24]. These models have demonstrated strong performance in similar tasks, and their hyperparameters have been extensively validated in the paper.

### A.4.3 Imputation Target Select Strategies

For training the denoising network, we randomly select some observed values as the imputation target. In this process, the mask ratio and mask pattern to get the imputation target directly determine the effectiveness of training. Thus, in this section, we explore the performance of FGTI with different mask ratios and mask patterns.

**Effect of Mask Ratio**   We first mask different ratios of observations as the imputation target for training, the performance of FGTI with different mask ratios is shown in Table 3. In addition, we consider a special case where observations with different ratios are randomly masked as imputation targets at each training step, instead of using a fixed masking ratio.

We can find that since the random ratio mask strategy can increase the learning complexity and enhance the modeling ability of the diffusion model, the random ratio mask strategy achieves optimal or sub-optimal performance in most cases. So we use the random ratio mask strategy by default.

**Effect of Mask Pattern**   Then we explore the performance when using different mask patterns. Following CSDI [44] and PriSTI [27], we consider three mask pattern strategies: (1) Block missing (2) Mix missing (3) Random missing. The results is shown in Table 4

Table 3: Varying the mask ratio of the imputation target when training the denoising network with 10% missing values

| Mask Ratio | KDD | | Guangzhou | | PhysioNet | |
| --- | --- | --- | --- | --- | --- | --- |
| | RMSE | MAE | RMSE | MAE | RMSE | MAE |
| 10% | 0.4372 | 0.1925 | 0.2388 | 0.1647 | 0.5992 | 0.3235 |
| 20% | 0.4143 | 0.1700 | **0.2312** | **0.1576** | 0.5853 | 0.2893 |
| 30% | 0.4257 | 0.1707 | 0.2335 | 0.1600 | 0.5836 | **0.2800** |
| 40% | 0.4185 | 0.1697 | 0.2372 | 0.1634 | 0.6181 | 0.3105 |
| 50% | 0.4183 | 0.1714 | 0.2343 | 0.1578 | 0.6308 | 0.3138 |
| Random Ratio | **0.4057** | **0.1489** | 0.2325 | 0.1584 | **0.5801** | 0.2856 |

Table 4: Varying the mask pattern of the imputation target when training the denoising network with 10% missing values

| Mask Pattern | KDD | | Guangzhou | | PhysioNet | |
| --- | --- | --- | --- | --- | --- | --- |
| | RMSE | MAE | RMSE | MAE | RMSE | MAE |
| Block missing | 0.4187 | 0.1778 | 0.2387 | 0.1596 | 0.6224 | 0.3384 |
| Mix missing | 0.4193 | 0.1792 | **0.2325** | **0.1531** | 0.6034 | 0.3208 |
| Random missing | **0.4057** | **0.1489** | **0.2325** | 0.1584 | **0.5801** | **0.2856** |

It can be found that Block missing or Mix missing strategy is not comparable to Random missing in most cases due to the possibility that the mask pattern may not correspond to the actual missing scenario. So we use the Random missing mask pattern by default.

## A.5 Comparative Experiments of Generative Baselines

To compare the imputation performance of FGTI with probabilistic generative baselines in more detail, we adopt CRPS [44] to evaluate the gap between the learned and ground truth distributions for different probabilistic generative methods following [44; 27].

First, we inject different rates of missing values by the MCAR mechanism, and report the CRPS performance with different missing rates in Table 5. Then we report the CRPS by varying the missing mechanism with 10% missing values in Table 6.

We can find that our method outperforms other probabilistic generative methods for all cases due to the introduction of frequency-domain conditions, thus providing empirical evidence for Proposition 3.1. It can be also found that the variations of CRPS are basically the same as RMSE and MAE for a specific model with different settings.

### A.5.1 Case Study

In order to verify the role of the high-frequency condition and the dominant-frequency condition, in this section we conduct a case study of FGTI for trend, seasonal, residual term over the pre-decomposed KDD dataset.

We first perform STL decomposition of the KDD dataset into Trend, Seasonal and Residual terms. Then we select 10% observations of the original KDD dataset as the mask positions by MCAR, and then mask the corresponding positions of the three terms. Finally we imputation the missing values in the three terms separately and report the performances. Note that this setup is the same as the survey experiment shown in Figure 1 in Section 1.

To study the role of high-frequency information, dominant-frequency information, and frequency-domain information, we consider the three ablation scenarios (1)w/o Dominant-frequency filter (2) w/o High-frequency filter and (3) w/o Frequency condition in Section 4.3.

We report the imputation results of different scenarios over different terms in Table 7

Table 5: CRPS of various probabilistic generative methods with different missing rates

| Dataset | Missing Rate | TimeCIB | GAIN | CSDI | SSSD | PriSTI | FGTI |
|---------|--------------|---------|------|------|------|--------|------|
| KDD | 10% | 0.466 | 0.709 | 0.224 | 0.352 | 0.232 | **0.158** |
| | 20% | 0.467 | 0.718 | 0.245 | 0.370 | 0.248 | **0.170** |
| | 30% | 0.469 | 0.729 | 0.259 | 0.374 | 0.268 | **0.186** |
| | 40% | 0.471 | 0.746 | 0.278 | 0.401 | 0.301 | **0.216** |
| Guang. | 10% | 0.360 | 0.692 | 0.265 | 0.316 | 0.209 | **0.155** |
| | 20% | 0.357 | 0.694 | 0.277 | 0.299 | 0.244 | **0.168** |
| | 30% | 0.356 | 0.695 | 0.292 | 0.353 | 0.310 | **0.193** |
| | 40% | 0.358 | 0.697 | 0.324 | 0.382 | 0.362 | **0.243** |
| Phy. | 10% | 0.466 | 0.739 | 0.544 | 0.617 | 0.444 | **0.343** |
| | 20% | 0.467 | 0.761 | 0.589 | 0.665 | 0.457 | **0.369** |
| | 30% | 0.469 | 0.787 | 0.627 | 0.630 | 0.467 | **0.389** |
| | 40% | 0.471 | 0.814 | 0.671 | 0.676 | 0.491 | **0.441** |

Table 6: CRPS of various probabilistic generative methods with different missing mechanisms

| Dataset | Miss mechanism | TimeCIB | GAIN | CSDI | SSSD | PriSTI | FGTI |
|---------|----------------|---------|------|------|------|--------|------|
| KDD | MCAR | 0.466 | 0.709 | 0.224 | 0.352 | 0.232 | **0.158** |
| | MAR | 0.470 | 0.710 | 0.229 | 0.489 | 0.239 | **0.164** |
| | MNAR | 0.490 | 0.715 | 0.244 | 0.456 | 0.252 | **0.174** |
| Guang. | MCAR | 0.360 | 0.692 | 0.265 | 0.316 | 0.209 | **0.155** |
| | MAR | 0.298 | 0.692 | 0.252 | 0.367 | 0.208 | **0.148** |
| | MNAR | 0.294 | 0.693 | 0.251 | 0.267 | 0.210 | **0.144** |
| Phy. | MCAR | 0.466 | 0.739 | 0.544 | 0.617 | 0.444 | **0.343** |
| | MAR | 0.593 | 0.739 | 0.550 | 0.724 | 0.454 | **0.356** |
| | MNAR | 0.608 | 0.743 | 0.566 | 0.715 | 0.476 | **0.366** |

We can find that for the trend term, retaining the dominant-frequency condition gives the best results, while the high-frequency condition may interfere with the imputation. For the seasonal term, the results are similar to the trend term, but the dominant-frequency information contributes less to the imputation for the seasonal term than for the trend term. This suggests that the seasonl term mainly corresponds to the dominant-frequency information, but also contains some of the high-frequency information. In contrast, the results of the experiments on the residual term show that the residual term mainly corresponds to high-frequency condition. Since we choose the transformer as the encoder in Cross-domain Representation Learning and utilize cross-attention as the fusion mechanism of the two frequency-domain conditions in time-frequency representation learning and attribute-frequency representation learning modules, our method can self-adaptively adjust the weights of the high-frequency information and the dominant-frequency information for different timestamps. Thus our method can outperform existing methods for datasets with multiple circumstances in most cases, as illustrated in Table 1.

### A.5.2 Visualizations

To showcase the imputation results of our FGTI model, we visualize the results with the state-of-the-art imputation methods CSDI and PriSTI in Figure 12 and Figure 13. We can find that the CSDI imputation results are not quite accurate for some fast-changing points due to the lack of sufficient condition guidance. On the other hand, both FGTI and PriSTI produced more accurate imputation results because they both used additional conditions. However, as shown in Tabel 1, FGTI still yields better results than PriSTI, suggesting that the high-frequency information and the dominant-frequency information we use are more superior to the interpolation information used by PriSTI.

### A.6 Societal Impact Statement

The development our FGTI imputation model could have a significant positive impact on various sectors including healthcare, finance, and environmental monitoring. In healthcare, improved imputation models can lead to more accurate health monitoring systems, enabling early detection and treatment

Table 7: Imputation results for the trend, seasonal and residual terms of KDD dataset with 10% missing values

| Component | Trend | | Seasonal | | Residual | |
|---|---|---|---|---|---|---|
| | RMSE | MAE | RMSE | MAE | RMSE | MAE |
| w/o Frequency condition | 0.0480 | 0.0155 | 0.0572 | 0.0364 | 0.5132 | 0.2975 |
| w/o Dominant-frequency filter | 0.0482 | 0.0157 | 0.0533 | 0.0334 | **0.4956** | **0.2814** |
| w/o High-frequency filter | **0.0409** | **0.0143** | **0.0485** | **0.0301** | 0.5129 | 0.2912 |
| FGTI | 0.0448 | 0.0159 | 0.0523 | 0.0325 | 0.5068 | 0.2885 |

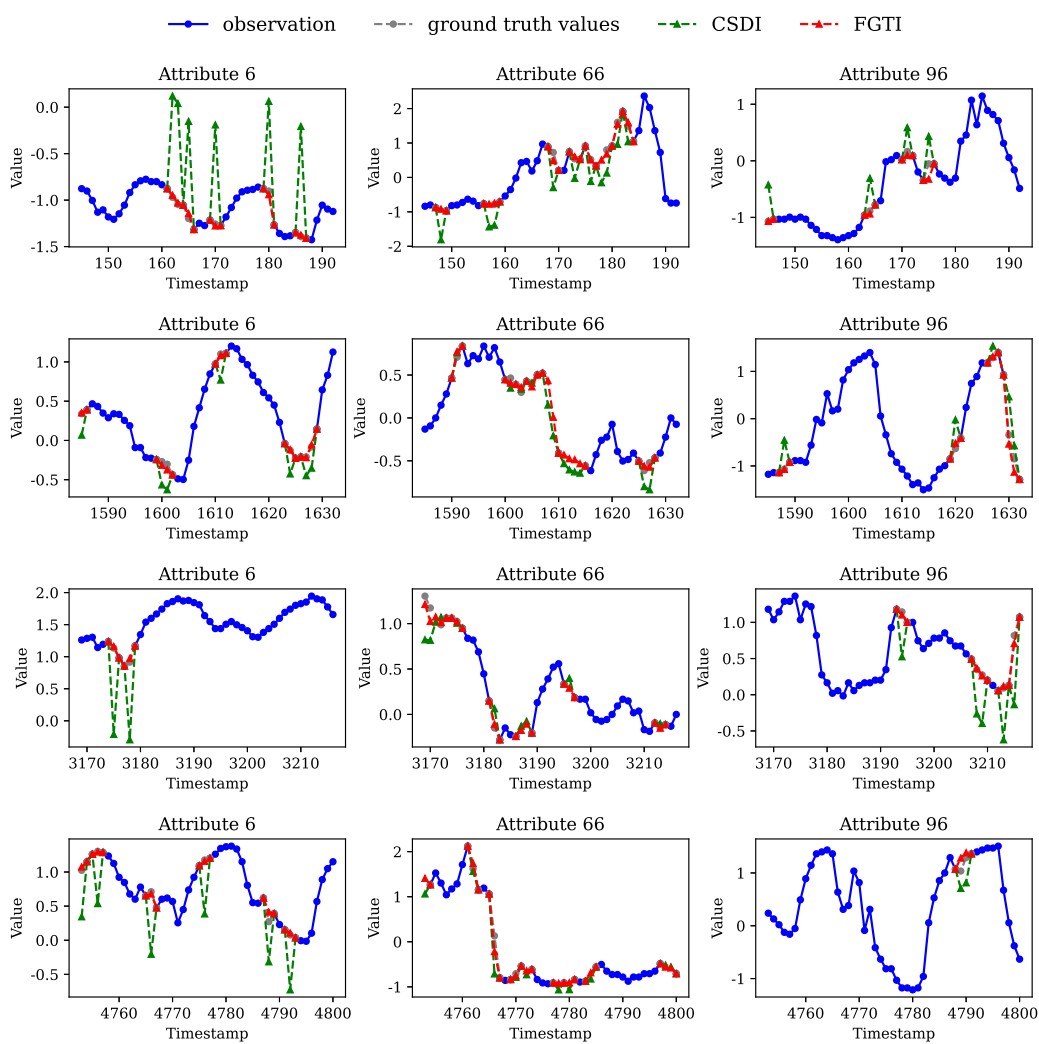

Figure 12: Imputation results visualization compared with CSDI for KDD dataset with 10% missing values.

of conditions by filling gaps in patient data. This can ultimately improve patient outcomes and reduce healthcare costs. In the financial sector, enhanced time series imputation can provide better forecasts and risk assessments, aiding in more informed decision-making and potentially stabilizing markets by decreasing uncertainty. Environmentally, better data imputation can improve weather prediction models and climate monitoring systems, aiding in disaster readiness and enhancing our ability to address climate change.

However, the deployment of advanced imputation models also raises certain concerns. If used in sensitive areas like surveillance, these models could lead to privacy invasions by reconstructing

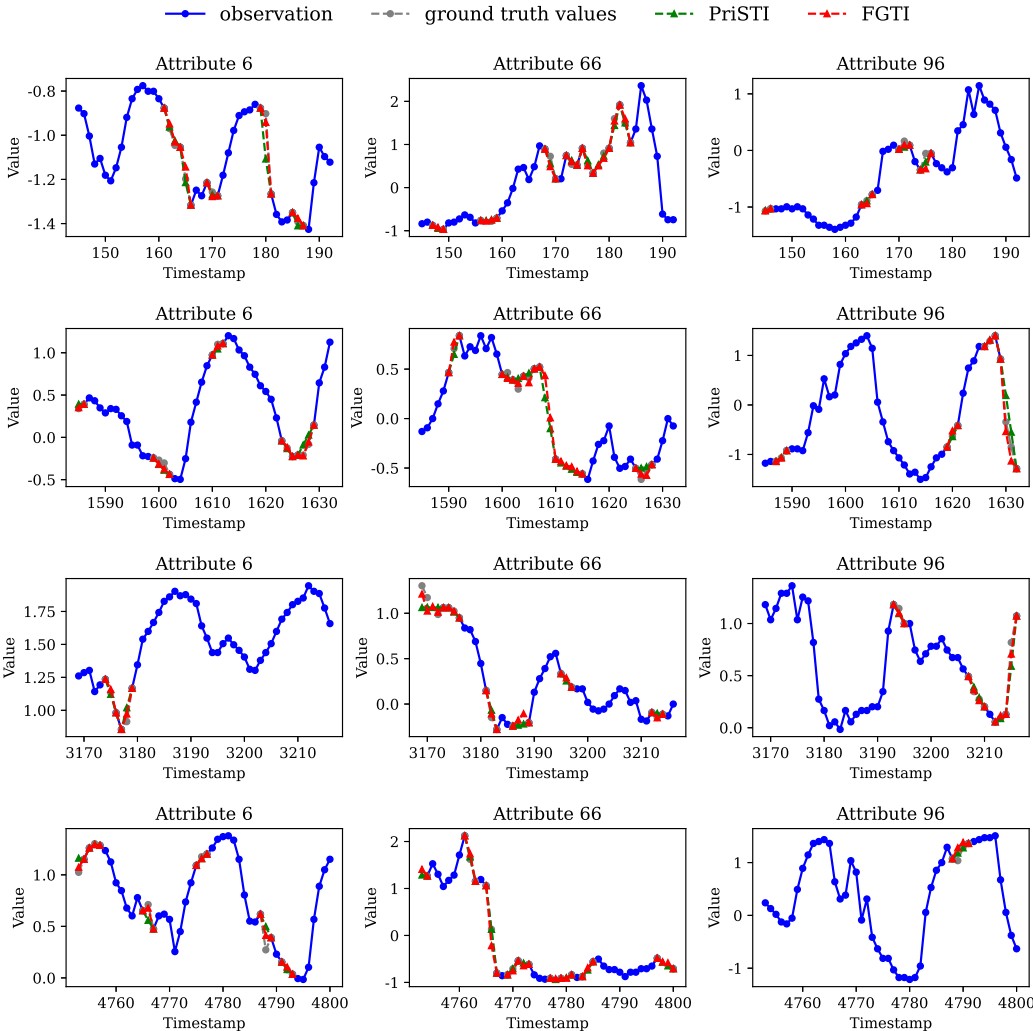

Figure 13: Imputation results visualization compared with PriSTI for KDD dataset with 10% missing values.

missing or incomplete data to track individuals without consent. In financial markets, sophisticated imputation methods could also exacerbate inequality by disproportionately benefiting institutions with the resources to leverage state-of-the-art technology, potentially leading to greater market dominance. Additionally, reliance on automated data imputation may result in complacency, where errors in imputation models propagate unnoticed, leading to decisions based on inaccurate or misleading data.

To mitigate negative impacts of our imputation model, implement stringent data privacy laws, and ethical guidelines, provide equal access to technology resources across entities, and establish rigorous validation processes to ensure accuracy and fairness. Regular auditing and transparency in algorithm deployment can also play a critical role.

