# OpenReview forum: "Frequency-aware Generative Models for Multivariate Time Series Imputation"
_NeurIPS.cc/2024/Conference — NeurIPS 2024 poster_

### Official Review · Reviewer_SUMs · 2024-07-08

**Soundness:** 3
**Presentation:** 2
**Contribution:** 3
**Rating:** 5
**Confidence:** 3

**Summary:**

This paper proposes FGTI that uses frequency-domain information with high-frequency and dominant-frequency filters for accurate imputation of residual, trend, and seasonal components.
Experimental results demonstrate FGTI's effectiveness in improving both data imputation accuracy and downstream applications.

**Strengths:**

1. The authors leverage high-frequency and dominant-frequency features to enhance time series imputation.

2. They introduce time-domain and frequency-domain representation learning modules to comprehensively capture relevant information.

3. Extensive experiments are conducted to demonstrate the method's effectiveness.

**Weaknesses:**

1. The formulation and writing of this paper should be improved, with attention to correcting several typos.

2. The authors claim that the residual of a time series mainly comprises high-frequency components. However, in many time series, there may be multiple scales of seasonality and trend features, potentially resulting in a final residual that is too noisy to contain useful information.

3. It appears that the proposed method utilizes frequency features of time series as conditions for the diffusion model, which seems straightforward and may limit its contribution.

**Questions:**

1. In Table 2, the absence of the High-frequency filter appears to impact performance less on KDD and Guangzhou datasets. However, on PhysioNet, the situation is reversed. What factors might explain this discrepancy?

2. How were the hyper-parameters for the experiments chosen? Could you elaborate on the methodology used for their selection?

3. What is the purpose of Proposition 3.1? It seems unnecessary and almost self-evident.

---

> ### Author Rebuttal · Authors · 2024-08-07
>
> Thank you for the thoughtful comments and valuable suggestions. We have tried our best to incorporate the suggestions in the revised version. Below, we provide our response to the questions and concerns.
>
> **1. (W1) The formulation and writing of this paper should be improved**
>
> We appreciate your feedback on improving clarity and grammatical accuracy. We have meticulously revised the manuscript, correcting errors such as "Recently, researchers attempt to utilize large language models" to "Recently, researchers have attempted to utilize large language models" (line 62), and ensuring all sentences are structurally sound and clearly presented.
>
> **2. （W2）Explain how multiple scales of seasonality and trend features in time series affect the residual’s usefulness, which may be too noisy**
>
> Thank you for your comments.
> Since our method does not explicitly perform STL decomposition, it is not affected by the multiple scales of trends and seasonality.
> Our method leverages Fourier transform to analyze both dominant-frequency and high-frequency information, which inherently addresses multiple scales of trend and seasonality.
> By our dominant-frequency filter, it is possible to capture multiple scales of trend and seasonal feature information.
> In addition, even when the high-frequency information that mainly contributes to the residual is noisy, we can still use the dominant-frequency information and adjust the weights of the high-frequency condition by attention layers for accurate imputation.
> As shown in Table 1, our method always outperforms other methods in different scenarios.
>
>
> **3. （W3）Straightforward and may limit its contribution**
>
> Our core insight is that existing methods cannot accurately imputation the residual term, therefore introducing the corresponding frequency-domain information.
> **The insight of modelling high-frequency signals for time series imputation has not yet been recognized by existing studies.**
> Our proposed filters and representation learning modules are simple and effective for imputation, and can shed light on further works.
> In addition, our proposed filters and representation learning modules can also be flexibly ported to other generative models and are not limited to applying diffusion models.
>
> **4. （Q1）Explain why the absence of the High-frequency filter impacts performance differently on KDD, Guangzhou, and PhysioNet datasets**
>
> Thank you for your question.
> The difference in impact of dominant-frequency and high-frequency information depends on the characteristics of datasets.
> For the PhysioNet dataset, the obtained dominant-frequency information may be biased due to it contains at least 79.71% missing values.
> For the KDD and Guangzhou datasets, the dominant-frequency information will play a larger role because of the lower missing rates.
> In these cases, our method can adaptively adjust the weight of high-frequency information and dominant-frequency information to help the imputation model.
>
> **5. （Q2）Hyper-parameters for the experiments chosen**
>
> For the two critical hyperparameters of the high-frequency filter and the dominant-frequency filter, we conduct empirical search to identify the optimal hyperparameters, as shown in Fig. 9 and Fig. 10 in Appendix A.4.2.
> For other settings related to diffusion model, we adopt hyperparameters recommended by the existing well-established models [23,43]. These models have demonstrated strong performance in similar tasks, and their hyperparameters have been extensively validated in those literature.
> To improve the presentation of the hyperparameters, we will add the above description of the parameters for the diffusion model to Appendix A.4.2.
>
> **6. （Q3）Purpose of Proposition**
>
> The purpose of Proposition 3.1 is to illustrate the mechanism behind introducing the high-frequency condition and the dominant frequency condition for facilitating generative models to impute missing values.
> This Proposition shows that with the introduction of the two conditions, the entropy of the target distribution will be reduced, the randomness of the target distribution will be narrowed, so that the generative models will more accurately impute missing values.
> To add empirical evidence for this proposition, we also add experiments using CRPS to evaluate the gap between the learned and ground truth distributions for different generative baselines in the following table:
>
> | Dataset | Miss. Rate | MIWAE  | GPVAE  | TimeCIB | GAIN   | CSDI   | SSSD   | PriSTI | FGTI   |
> |---------|------------|--------|--------|---------|--------|--------|--------|--------|--------|
> | KDD     | 10%        | 0.524  | 0.443  | 0.466   | 0.709  | 0.224  | 0.352  | 0.232  | **0.158**  |
> |         | 20%        | 0.526  | 0.445  | 0.467   | 0.718  | 0.245  | 0.370  | 0.248  | **0.170**  |
> |         | 30%        | 0.532  | 0.447  | 0.469   | 0.729  | 0.259  | 0.374  | 0.268  | **0.186**  |
> |         | 40%        | 0.530  | 0.457  | 0.471   | 0.746  | 0.278  | 0.401  | 0.301  | **0.216**  |
> | Guang.  | 10%        | 0.312  | 0.333  | 0.360   | 0.692  | 0.265  | 0.316  | 0.209  | **0.155**  |
> |         | 20%        | 0.312  | 0.330  | 0.357   | 0.694  | 0.277  | 0.299  | 0.244  | **0.168**  |
> |         | 30%        | 0.312  | 0.335  | 0.356   | 0.695  | 0.292  | 0.353  | 0.310  | **0.193**  |
> |         | 40%        | 0.312  | 0.333  | 0.358   | 0.697  | 0.324  | 0.382  | 0.362  | **0.243**  |
> | Phy.    | 10%        | 0.689  | 0.659  | 0.466   | 0.739  | 0.544  | 0.617  | 0.444  | **0.343**  |
> |         | 20%        | 0.717  | 0.665  | 0.467   | 0.761  | 0.589  | 0.665  | 0.457  | **0.369**  |
> |         | 30%        | 0.750  | 0.674  | 0.469   | 0.787  | 0.627  | 0.630  | 0.467  | **0.389**  |
> |         | 40%        | 0.779  | 0.680  | 0.471   | 0.814  | 0.671  | 0.676  | 0.491  | **0.441**  |
>
> As shown in the above table and Table 1, our method outperforms other methods for all metrics due to the introduction of frequency domain conditions, thus echoing the proposition.

---

### Official Review · Reviewer_Q2t8 · 2024-07-13

**Soundness:** 3
**Presentation:** 3
**Contribution:** 3
**Rating:** 5
**Confidence:** 3

**Summary:**

This paper proposes a generative model, Frequency-aware Generative Models for Multivariate Time Series Imputation (FGTI), for multivariate time series imputation. The proposed model is designed to enhance the imputation performance by modeling the residual component of time series data. To this end, the paper leverages the observation that the residual components are usually high-frequent. It utilizes two filters in the frequency domain to extract high-frequency and dominant-frequency information from the time series using FFT and IFFT. The frequency information is then input into a denoising network to generate the missing data conditioned on the observed part of the time series. Experiments are performed using three real-world datasets and a number of recent baselines are benchmarked.

**Strengths:**

- The idea of using frequency filters to model the residual component of time-series data is natural and reasonable.
- The empirical evaluation is systematic and the experiment results show consistent improvement compared with existing methods.

**Weaknesses:**

- It is not clear if the modeling of temporal dependency is sufficient in the time-frequency and attribute-frequency representation learning module. From Eq. (8-9), it seems that only two attention layers are used without considering the time stamp.
- The model architecture and specification used for Projectors in the denoising network is not described.
- There is a gap between Sections 3.2 and 3.3. The time-frequency representation and attribute-frequency representation learning are described and the final output $\mathbf{R}^a$ is defined. However, it is not clear how this representation is used in the diffusion model. It seems from Eq. (14) and Appendix A.2.1 that $\mathbf{R}^a$ is not involved in the parameterization $\epsilon_\theta$.

**Questions:**

Please kindly see and clarify my comments in the Weaknesses section.

**Limitations:**

Limitations of high demand of computational resources are discusses with relevant experiment results analyzed.

---

> ### Author Rebuttal · Authors · 2024-08-07
>
> We really appreciate your thoughtful comments and valuable suggestions. We have incorporated the suggestions in the revised paper. Below, we provide our response to the concerns.
>
> **1. (W1) Clarify if temporal dependency modelling is sufficient**
>
> We would really appreciate your feedback.
> We apologize for the inadequate illustrations and explanations of the time-frequency and attribute-frequency representation learning modules
> In line 144, we state that the Encoder(.) is implemented by the transformer backbone.
> Since a position encoding layer exists in this encoder, we can add the timestamp information for the corresponding Query and Key in the two representation learning modules.
> To improve the presentation, **we add a figure illustrating the detailed architecture of the denoising network, which is uploaded in the supplementary pdf file at the top of the page**.
> And we will add this figure to Section 3.3.
> In addition, the setup where modelling temporal dependencies than modelling attribute dependencies by two attention layers is common in time series imputation methods [26,43].
> Considering that our method outperforms other imputation methods for different missing rates and different missing mechanisms in Table 1, Fig. 4, Fig. 7 and Fig.8, the modelling of the time dependence is sufficient.
>
> **2. (W2) Describe the model architecture and specifications used for Projectors in the denoising network**
>
> Thank you for your suggestions.
> We apologize for the inadequate description and explanation of the detailed structure of the denoising network.
> As shown in Fig.2, which is uploaded in the supplemental pdf, the input projector is an MLP layer, and the output projector is a 2-layer MLP with a ReLU activation function.
> This setup is consistent with the other SOTA imputation methods [26,43], which have been extensively validated in those literature.
>
> **3. (W3) Clarify how time-frequency and attribute-frequency representations are used in the diffusion model**
>
> Thank you for your valuable suggestions for boosting our manuscript.
> We apologise for not having a very clear representation of the denoising network architecture of the diffusion model.
> The Fig.2 in supplemental pdf illustrating the detailed architecture of the denoising network will add in the Section 3.3.
> To achieve the imputation of generative models that are helped by the guidance of frequency-domain information, we should incorporate the representation learning modules into the generative model architecture.
> As shown in the figure, $\mathbf{R}^{a}$ is a hidden vector in the denoising neural network of the diffusion model.

---

> > ### Comment · Reviewer_Q2t8 · 2024-08-10
> > **Thank you for your response**
> >
> > I appreaciate the response from the authors and the additional figure to clarify the model architecture. My concerns have been addressed. Specifically,
> > - The time information can indeed by captured by Transformer models. I would suggest authors consider spelling this out by revising Eq. (7) from $\text{Encoder}(\cdot)$ to $\text{Transformer}(\cdot)$ to enhance clarity.
> > - The architecture and specifications for the Projectors are clarified. Please add the relevant descriptions in the main text.
> > - The use of $\mathbf{R}^a$ and $\mathbf{R}^t$ are clarified with the revised architecture figure.
> >
> > Given that my concerns are addressed, I increase my overall rating from 4 to 5.

---

> > > ### Author Response · Authors · 2024-08-10
> > > **Thanks for the response**
> > >
> > > Thank you for your thoughtful evaluation and the feedback provided in your review. We are pleased to hear that the additional figure and our responses have helped in addressing your concerns.
> > >
> > > - We will certainly revise Equation (7) and Fig.3 to replace Encoder() with Transformer() to more explicitly indicate that time information can be captured via Transformer models, as you've suggested.
> > >
> > > - Additionally, we will incorporate detailed descriptions of the input projector and output projector in the main text,
> > > as detailed in the Rebuttal of response W2,
> > > to avoid any ambiguities regarding their architecture and specifications.
> > >
> > > - We also acknowledge your satisfaction with the clarification provided through the revised figure regarding the use of
> > > Ra and Rt. We will ensure that similar clarity is maintained throughout the manuscript to aid the understanding of our model components.
> > >
> > > Thank you again for your invaluable feedback and guidance. If you have any questions, please send them to us, we look forward to discussing with you to further improve our work.
> > >
> > > Best regards,
> > >
> > > Authors

---

### Official Review · Reviewer_qcDA · 2024-07-16

**Soundness:** 4
**Presentation:** 4
**Contribution:** 3
**Rating:** 6
**Confidence:** 4

**Summary:**

The paper proposes a new model, called FGTI, to address the issue of missing data in multivariate time series extracting frequency-domain information. The authors argue that existing methods, in general, neglect the residual term, which is the most significant contributor to imputation errors. FGTI incorporates frequency-domain information using high-frequency and dominant-frequency filters to improve the imputation of the residual, trend, and seasonal terms. The model is evaluated on three real-world datasets and demonstrates superior performance in both imputation accuracy and downstream tasks compared to existing methods.

**Strengths:**

- The paper is well-written and well-motivated.
- The authors tackle a very important problem of time-series imputation, focusing on fully utilizing frequency-domain information, which has not been considered enough.
- The experiments are thorough, including ablation studies and different missing scenarios. The method is compared with cutting-edge baselines.

**Weaknesses:**

- The experiments are thorough and include multiple state-of-the-art imputation methods. However, the paper misses two important time-series imputation methods: one with a VAE-based approach [A] and one applying trend-seasonality decomposition [B]. While it's understandable that the authors might not have had enough time to include these papers since they are from ICLR 2024, technical comparisons with these methods would be necessary before acceptance.

- While the authors provided experiments with various missing scenarios and missing rates, it is not clear why the proposed method outperforms the conventional methods and under what circumstances. For instance, extracting high-frequency information could be significantly distorted if missingness occurs with a certain frequency (which seems realistic considering periodic noises). The paper would be more convincing if the authors provided experiments that assess the effect of the trend and seasonality components of the datasets (or using synthetic datasets) and missing patterns.

- The experiments would be more appealing if the authors provided a breakdown (seasonality and trend) of the assessments on the imputed time-series.

Minor comment:

- Regarding Figure 1, since the RMSE and MAE comparison essentially conveys similar information about the population-level importance of the trend-seasonality decomposition, replacing one with specific time-series examples where the trend and seasonality of the missing values are correctly imputed would better illustrate the motivation. Also, information about how Figure 1 is reported is missing.

References:

[A] Choi and Lee, "Conditional Information Bottleneck Approach for Time Series Imputation, ICLR 2024.
[B] Liu et al., "Multivariate Time-series Imputation with Disentangled Temporal Representations," ICLR 2024.

**Questions:**

- Regarding Weakness 2: Please describe why the proposed method works better than conventional methods and under what circumstances. Do the seasonality and trend profiles of the given time-series dataset matter, and if so, how?

- It seems that how to mask the input (to learn the underlying diffusion model) is very important. How does the model's performance change depending on the masking ratio and/or masking patterns? This is crucial as masking may distort the trend and seasonality components of the given time-series.

---

> ### Author Rebuttal · Authors · 2024-08-07
>
> We thank the constructive comments and suggestions for further experiments. We will try our best to incorporate the suggestions in the revised version.
> Below, we provide our response to the questions and concerns.
>
> **1. (W1)  Compared with the most recent baselines**
>
> Thank you for the valuable suggestions.
> TimeCIB [a] is a very competitive VAE-based imputation method, so we incorporate TimeCIB into the baselines for comparative experiments.
> In addition, we have already considered TIDER [27] in our experiments.
>
> We first report the part of updated Table 1 for the added baseline:
> | 　     | Miss. Rate | Metric |  FreTS |    FGTI    |
> |--------|------------|--------|:------:|:----------:|
> | KDD    | 10%        | RMSE   | 0.630  | **0.406** |
> |        | 　         | MAE    | 0.412  | **0.149** |
> |        | 20%        | RMSE   | 0.741  | **0.451** |
> |        | 　         | MAE    | 0.489  | **0.161** |
> |        | 30%        | RMSE   | 0.796  | **0.448** |
> |        | 　         | MAE    | 0.546  | **0.176** |
> |        | 40%        | RMSE   | 0.850  | **0.478** |
> | 　     | 　         | MAE    | 0.591  | **0.205** |
> | Guang. | 10%        | RMSE   | 0.456  | **0.230** |
> |        | 　         | MAE    | 0.340  | **0.170** |
> |        | 20%        | RMSE   | 0.602  | **0.258** |
> |        | 　         | MAE    | 0.460  | **0.176** |
> |        | 30%        | RMSE   | 0.709  | **0.291** |
> |        | 　         | MAE    | 0.547  | **0.202** |
> |        | 40%        | RMSE   | 0.787  | **0.356** |
> | 　     | 　         | MAE    | 0.611  | **0.254** |
> | Phy.   | 10%        | RMSE   | 0.804  | **0.580** |
> |        | 　         | MAE    | 0.540  | **0.286** |
> |        | 20%        | RMSE   | 0.825  | **0.577** |
> |        | 　         | MAE    | 0.576  | **0.309** |
> |        | 30%        | RMSE   | 0.861  | **0.624** |
> |        | 　         | MAE    | 0.603  | **0.336** |
> |        | 40%        | RMSE   | 0.883  | **0.669** |
> | 　     | 　         | MAE    | 0.626  | **0.376** |
>
> We then report the imputation results for varying missing mechanism with 10% missing values in the following table, and update Fig.7, Fig.8 and Fig.9:
> | 　     | Miss. Mech. | Metric | TimeCIB | FGTI       |
> |--------|-------------|--------|---------|------------|
> | KDD    | MCAR        | RMSE   | 0.589   | **0.406** |
> |        | 　          | MAE    | 0.367   | **0.149** |
> |        | MAR         | RMSE   | 0.589   | **0.406** |
> |        | 　          | MAE    | 0.367   | **0.149** |
> |        | MNAR        | RMSE   | 0.693   | **0.499** |
> |        | 　          | MAE    | 0.408   | **0.174** |
> | Guang. | MCAR        | RMSE   | 0.451   | **0.230** |
> |        | 　          | MAE    | 0.300   | **0.170** |
> |        | MAR         | RMSE   | 0.376   | **0.218** |
> |        | 　          | MAE    | 0.245   | **0.150** |
> |        | MNAR        | RMSE   | 0.327   | **0.200** |
> |        | 　          | MAE    | 0.230   | **0.140** |
> | Phy.   | MCAR        | RMSE   | 0.697   | **0.580** |
> |        | 　          | MAE    | 0.450   | **0.286** |
> |        | MAR         | RMSE   | 0.327   | **0.200** |
> |        | 　          | MAE    | 0.230   | **0.140** |
> |        | MNAR        | RMSE   | 0.697   | **0.580** |
> |        | 　          | MAE    | 0.450   | **0.286** |
>
> After this, we report the updated resource consumption over the KDD dataset with 10% missing values in the following table, and update Fig.5:
> |          | GPU Consumption Usage (MiB) | Running Time (s) |
> |----------|-----------------------------|------------------|
> | TimeCIB  | 930                         | 6.52             |
> | FGTI     | 9103                        | 3874.30          |
>
> Finally, we report on the updated downstream application experiments in the following table,
> and update Fig.6:
> |            | Air quality prediction (RMSE) | Mortality forecast (AUC) |
> |------------|-------------------------------|--------------------------|
> | TimeCIB    | 0.63                          | 0.82                     |
> | FGTI       | **0.59**                      | **0.86**                 |
>
> We can find that FGTI still outperforms all baselines due to the guidance of high-frequency information and dominant-frequency information, as well as the generating ability of the diffusion model.

---

> ### Author Response · Authors · 2024-08-07
> **Continued response for W2, Q1**
>
> **2. (W2,Q1) Clarify why your method outperforms conventional ones, considering trend, seasonality, and missing patterns in datasets**
>
>  Thank you for your valuable suggestions for boosting our manuscript.
> We conduct a case study of FGTI for trend, seasonal, and residual terms over the pre-decomposed KDD dataset.
> We first perform STL decomposition of the KDD dataset into Trend, Seasonal and Residual terms.
> Then we select 10% observations of the original KDD dataset as the mask positions by MCAR,
> and then mask the corresponding positions of the three terms.
> To study the role of high-frequency information, dominant-frequency information,
> and frequency-domain information,
> we consider the three ablation scenarios: (1) w/o Dominant-frequency filter
> (2) w/o High-frequency filter and (3) w/o Frequency condition in Section 4.3.
> We report the imputation results of different scenarios over different terms in following table:
> | Component                     | Trend      |            | Seasonal   |            | Residual   |            |
> |-------------------------------|------------|------------|------------|------------|------------|------------|
> |                               | RMSE       | MAE        | RMSE       | MAE        | RMSE       | MAE        |
> | w/o Frequency condition       | 0.048      | 0.0155     | 0.0572     | 0.0364     | 0.5132     | 0.2975     |
> | w/o Dominant-frequency filter | 0.0482     | 0.0157     | 0.0533     | 0.0334     | **0.4956** | **0.2814** |
> | w/o High-frequency filter     | **0.0409** | **0.0143** | **0.0485** | **0.0301** | 0.5129     | 0.2912     |
> | FGTI                          | 0.0448     | 0.0159     | 0.0523     | 0.0325     | 0.5068     | 0.2885     |
>
> We can find that for the Trend term, retaining the dominant-frequency information gives the best results,
> while the-high frequency information may interfere the imputation.
> For the Seasonal term, the results are similar to the Trend term,
> but the dominant-frequency information contributes less to the imputation for the Seasonal term than for the Trend term.
> This suggests that the Seasonl term mainly corresponds to the dominant-frequency information, but also contains some of the high-frequency information.
> In contrast, the results of the experiments on the Residual term show that it mainly corresponds to high-frequency information.
> Since we choose Transformer as the encoder in Cross-domain Representation Learning and utilize cross-attention as the fusion mechanism of the two frequency domain information in Time-frequency Representation Learning and Attribute-frequency Representation Learning, our method can self-adaptively adjust the weights of the high-frequency information and the dominant-frequency information for different timestamps.
> Thus our method can outperform conventional methods for datasets with any circumstances in most cases, as illustrated in Table 1.
>
> In addition, our method's pipeline does not perform STL decomposition of the time series, but extracts the frequency domain informations to guide imputation, so we do not need to know the trend and seasonality of the dataset.
> Thanks to the encoder structure and cross-attention based fusion mechanism in Cross-domain Representation Learning, we can adaptively adjust the weights of the two frequency domain information on datasets dominated by different terms.
>
> Furthermore, we consider three missing mechanisms, MCAR, MAR, and MNAR, in our comparative experiments. MCAR implies that each attribute could be missing at any frequency, while MAR potentially implies that all attributes' missing status depends on the frequency of a particular attribute (e.g., high temperatures could cause sensor failure).
> For MNAR, the missing status of all attributes depends on the frequency of that attribute.
> As shown in Fig 4, Fig 7 and Fig 8 in the manuscript, our method still outperforms conventional methods
> with different missing mechanisms (patterns).

---

> ### Author Response · Authors · 2024-08-07
> **Continued response for W3, C1, Q2**
>
> **3. (W3) Provided a breakdown (seasonality and trend) of the assessments**
>
> Thank you for your suggestions.
> The pipeline of our method does not perform a direct STL decomposition, but instead uses the high-frequency and dominant-frequency information to potentially guide the imputation of the three terms.
> Due to the effect of missing values, we will get different decomposition results before and after imputation, so for each term the ground truth will be lacking for the evaluation.
> Therefore, we first perform STL decomposition of the original KDD dataset, then inject the missing values for the three terms respectively to verify the imputation results, and the results are shown in Fig. 1 of Section 1.
> In addition, we also add a Case study about the trend, seasonal and residual terms using the pre-decomposed KDD dataset, and the results are shown in the above table.
> The results show that dominant-frequency information can mainly help the imptuation of the Trend and Seasonal terms, and the high-frequency information is mainly help to the Residual term.
>
> **4. (C1) Replace RMSE or MAE with examples, and clarify Figure 1 reporting details**
>
> Thank you for your valuable suggestions for boosting our manuscript.
> Regarding the setting for the study of Fig. 1, we first perform STL decomposition of the KDD dataset into Trend, Seasonal and Residual terms.
> Then we select 10% observations of the original KDD dataset as the mask positions by MCAR, and mask the corresponding positions of the three terms.
> Finally, we impute the missing values of the three terms by different methods separately.
> Following your valuable suggestion, **we add the examples of imputations on trend, seasonal and residual terms in Fig. 1, which is uploaded in the supplementary pdf file at the top of the page**.
> and add the setup for this survey study.
> The results also indicate that the SOTA time-series imputation methods are inaccurate in imputing the residual term.
>
> **5. (Q2) Explain how masking ratio/patterns affect model performance**
>
> Thank you for your comments. We agree that mask ratio and mask pattern are critical for the learning process.
> We randomly mask different ratios of observations as the imputation target at each training step, instead of using a fixed mask ratio, following CSDI [43].
> The performance of FGTI with different mask ratios is shown in the following table:
> | Mask Ratio | KDD     |         | Guangzhou |         | PhysioNet |         |
> |------------|---------|---------|-----------|---------|-----------|---------|
> |            | RMSE    | MAE     | RMSE      | MAE     | RMSE      | MAE     |
> | 10%        | 0.4372  | 0.1925  | 0.2388    | 0.1647  | 0.5992    | 0.3235  |
> | 20%        | 0.4143  | 0.1700  | **0.2312**    | **0.1576**  | 0.5853    | **0.2893**  |
> | 30%        | 0.4257  | 0.1707  | 0.2335    | 0.1600  | 0.5836    | 0.2800  |
> | 40%        | 0.4185  | 0.1697  | 0.2372    | 0.1634  | 0.6181    | 0.3105  |
> | 50%        | 0.4183  | 0.1714  | 0.2343    | 0.1578  | 0.6308    | 0.3138  |
> | Random     | **0.4057**  | **0.1489**  | 0.2325    | 0.1584  | **0.5801**    | 0.2856  |
>
> We can find that since the random mask strategy can increase the learning complexity and enhance the modelling ability of the diffusion model, thus our masking strategy achieves optimal or sub-optimal performance in most cases.
>
> Then, we explore the performance when using different mask patterns.
> Following CSDI [43] and PriSTI [26], we consider (1) Block missing (2) Mix missing strategy (3) Random missing mask pattern，the results is shown in the following table:
> | Mask Pattern | KDD     |         | Guangzhou |         | PhysioNet |         |
> |--------------|---------|---------|-----------|---------|-----------|---------|
> |              | RMSE    | MAE     | RMSE      | MAE     | RMSE      | MAE     |
> | Block        | 0.4187  | 0.1778  | 0.2387    | 0.1596  | 0.6224    | 0.3384  |
> | Mix          | 0.4193  | 0.1792  | **0.2325**    | **0.1531**  | 0.6034    | 0.3208  |
> | Random       | **0.4057**  | **0.1489**  | **0.2325**    | 0.1584  | **0.5801**    | **0.2856**  |
>
> It can be found that Block missing or Mix missing is not comparable to Random missing in most cases due to the possibility that the mask pattern may not correspond to the actual missing scenario.
> So, we use the Random missing mask pattern by default.
>
> Reference:
>
> [a] Conditional Information Bottleneck Approach for Time Series Imputation

---

> > ### Comment · Reviewer_qcDA · 2024-08-11
> >
> > I have read all the rebuttals and I fully appreciate the experiments the authors made. This is an interesting paper with good motivation and thorough exepriments. I tend to keep my initial decision, which is "Weak Accept".

---

> > > ### Author Response · Authors · 2024-08-11
> > > **Thanks for the response**
> > >
> > > Thank you for appreciating our detailed rebuttals and experimental efforts. We are delighted to know you find our paper interesting with solid motivation and thorough experiments.
> > >
> > > **Given these positive remarks and to strengthen our position for final acceptance, may we kindly request you to reconsider your score and potentially upgrade it?**  This would greatly enhance our chances of contributing our findings to a broader audience.
> > >
> > > Best regards,
> > >
> > > Authors

---

### Official Review · Reviewer_WWjA · 2024-07-21

**Soundness:** 3
**Presentation:** 3
**Contribution:** 3
**Rating:** 5
**Confidence:** 4

**Summary:**

The authors present Frequency-aware Generative Models for Multivariate Time Series Imputation (FGTI), a model which addresses the challenge of missing data in multivariate time series by focusing on the often-overlooked residual term. The paper also incorporates frequency-domain information to enhance imputation performance. Experiments show that this outperforms many existing time series imputation baselines on three real-world datasets.

**Strengths:**

S1. The work proposes an approach for time series imputation, which is an important topic for a wide range of applications.

S2. The authors aim to address the influence of frequency-domain information, w.r.t, both high-frequency condition and dominant-domain condition.

S3. Extensive experiments have been carried out.

**Weaknesses:**

W1. In Introduction and Section 3.1.2, it shows less explanation on why frequency components with large amplitudes could guide both the imputation of trend and seasonal terms. Did it address the intricately entangled trend-seasonal representations, which is highly important in current approaches?

W2. In the section of Related Work, I did not find discussions regarding existing research work and this study in frequency domain for time series tasks, this would be hard to clearly justify the main difference/novelty of this paper compared with current approaches.

W3. In the evaluation, only RMSE and MAE are used as metrics. However, it would be better to include additional metrics such as CRPS.

W4. It would be fairer if the authors compared their approach with a time series model using frequency domain information, which also addresses the trend, seasonal, and residual terms.

W5. Both CSDI and PriSTI address the issue of missing strategies. From the results of Figure 4, 7 and 8, the improvement of MAR and MNAR was much lower than that of MCAR on the Guangzhou and Physio datasets. Was it due to differences in the data distribution, or was it due to the setting of the hyper-parameters?

[1] Learning Latent Seasonal-Trend Representations for Time Series Forecasting.
[2] Frequency-domain MLPs are More Effective Learners in Time Series Forecasting
[3] CSDI: Conditional Score-based Diffusion Models for Probabilistic Time Series Imputation
[4] PriSTI: A Conditional Diffusion Framework for Spatiotemporal Imputation

**Questions:**

Please see Weaknesses.

**Limitations:**

1.	My main concern with the work is that the contribution of the proposed method might be incremental. Compared with CSDI, frequency domain filters are utilized to incorporate frequency domain information, and further cross-domain representation learning and frequency-aware diffusion framework are natural implementations, which lack additional novelty.

---

> ### Author Rebuttal · Authors · 2024-08-07
>
> Thank you for the thoughtful comments and valuable suggestions. Below, we provide our response to the questions and concerns.
>
> **1. (W1) Why large amplitude frequency components guide trend and seasonal term imputation**
>
> Thank you for your comments.
> We recognise that our current expression is insufficient.
> In STL decomposition, the dominant frequency components(with large amlitude) typically correspond to the trend and seasonal terms, while the high-frequency information is often associated with the residual term [a].
> This is because trend captures long-term movements, and seasonal captures major period patterns.
> We also add an empirical evidence on the pre-decomposed KDD dataset in following table:
> | Component                     | Trend      |            | Seasonal   |            | Residual   |            |
> |-------------------------------|------------|------------|------------|------------|------------|------------|
> |                               | RMSE       | MAE        | RMSE       | MAE        | RMSE       | MAE        |
> | w/o Frequency condition       | 0.048      | 0.0155     | 0.0572     | 0.0364     | 0.5132     | 0.2975     |
> | w/o Dominant-frequency filter （Preserving high-frequency information） | 0.0482     | 0.0157     | 0.0533     | 0.0334     | **0.4956** | **0.2814** |
> | w/o High-frequency filter （Preserving dominant-frequency information）    | **0.0409** | **0.0143** | **0.0485** | **0.0301** | 0.5129     | 0.2912     |
> | FGTI                          | 0.0448     | 0.0159     | 0.0523     | 0.0325     | 0.5068     | 0.2885     |
>
> This suggests that the Trend term mainly corresponds to the dominant-frequency information, the Seasonl term mainly corresponds to the dominant-frequency information, but also contains some of the high-frequency information, the Residual term mainly corresponds to high-frequency information.
> Since we choose transformer as the encoder and utilize cross-attention as the fusion mechanism of the two frequency domain information, our method can self-adaptively adjust the weights of the high-frequency information and the dominant-frequency information for the characteristics of the three terms in different datasets.
>
> And we will add the content of the above expressions in Section.1 and Section 3.1.2.
>
> **2. (W2) Discuss existing frequency domain research for time series tasks**
>
> Thank you for your valuable suggestions for improving the quality of the manuscript.
> Our current Related work section mainly focuses on the currently state-of-the-art time series imputaiton methods.
> For the time series imputation methods in frequency domain, mvLSWimpute [b] utilizes wavelet transforms to guide imputation, APDNet [c] uses the Fourier Temporal and Fourier Variable Interaction modules to model dependencies.
> In addition, the frequency domain time series forecasting methods FEDformer [d], FreTS [e] can also be applied to imputation task.
> However, they did not consider how to use frequency domain information to accurately model the residual terms of the missing data,
> which is critical for boosting the overall imputation performance.
> In contrast, our FGTI captures high-frequency information and dominant-frequency information to get a more accurate modeling of the residual term, while assisting in describing trend and seasonal terms.
>
> We will add to the discussion on imputation methods in frequency domain to the related work section.

---

> ### Author Response · Authors · 2024-08-07
> **Continued response for W3**
>
> **3. (W3) Include additional metrics such as CRPS**
>
> Thank you for the thoughtful comments and valuable suggestions.
> Since the CRPS is used to evaluate probabilistic imputation baselines, we report the CRPS for all probabilistic imputation baselines  according to the experimental settings in CSDI [43]and PriSTI [26].
> First, we report the CRPS performance with different missing rates in the following table：
> | Dataset | Miss. Rate | MIWAE  | GPVAE  | TimeCIB | GAIN   | CSDI   | SSSD   | PriSTI | FGTI   |
> |---------|------------|--------|--------|---------|--------|--------|--------|--------|--------|
> | KDD     | 10%        | 0.524  | 0.443  | 0.466   | 0.709  | 0.224  | 0.352  | 0.232  | **0.158**  |
> |         | 20%        | 0.526  | 0.445  | 0.467   | 0.718  | 0.245  | 0.370  | 0.248  | **0.170**  |
> |         | 30%        | 0.532  | 0.447  | 0.469   | 0.729  | 0.259  | 0.374  | 0.268  | **0.186**  |
> |         | 40%        | 0.530  | 0.457  | 0.471   | 0.746  | 0.278  | 0.401  | 0.301  | **0.216**  |
> | Guang.  | 10%        | 0.312  | 0.333  | 0.360   | 0.692  | 0.265  | 0.316  | 0.209  | **0.155**  |
> |         | 20%        | 0.312  | 0.330  | 0.357   | 0.694  | 0.277  | 0.299  | 0.244  | **0.168**  |
> |         | 30%        | 0.312  | 0.335  | 0.356   | 0.695  | 0.292  | 0.353  | 0.310  | **0.193**  |
> |         | 40%        | 0.312  | 0.333  | 0.358   | 0.697  | 0.324  | 0.382  | 0.362  | **0.243**  |
> | Phy.    | 10%        | 0.689  | 0.659  | 0.466   | 0.739  | 0.544  | 0.617  | 0.444  | **0.343**  |
> |         | 20%        | 0.717  | 0.665  | 0.467   | 0.761  | 0.589  | 0.665  | 0.457  | **0.369**  |
> |         | 30%        | 0.750  | 0.674  | 0.469   | 0.787  | 0.627  | 0.630  | 0.467  | **0.389**  |
> |         | 40%        | 0.779  | 0.680  | 0.471   | 0.814  | 0.671  | 0.676  | 0.491  | **0.441**  |
>
> Then we report the CRPS by varying the missing mechanism with 10% missing values in the following table：
> | Dataset | Miss. Mech. | MIWAE  | GPVAE  | TimeCIB | GAIN   | CSDI   | SSSD   | PriSTI | FGTI   |
> |---------|-------------|--------|--------|---------|--------|--------|--------|--------|--------|
> | KDD     | MCAR        | 0.524  | 0.443  | 0.466   | 0.709  | 0.224  | 0.352  | 0.232  | **0.158**  |
> |         | MAR         | 0.539  | 0.437  | 0.470   | 0.710  | 0.229  | 0.489  | 0.239  | **0.164**  |
> |         | MNAR        | 0.615  | 0.435  | 0.490   | 0.715  | 0.244  | 0.456  | 0.252  | **0.174**  |
> | Guang.  | MCAR        | 0.312  | 0.333  | 0.360   | 0.692  | 0.265  | 0.316  | 0.209  | **0.155**  |
> |         | MAR         | 0.258  | 0.334  | 0.298   | 0.692  | 0.252  | 0.367  | 0.208  | **0.148**  |
> |         | MNAR        | 0.241  | 0.340  | 0.294   | 0.693  | 0.251  | 0.267  | 0.210  | **0.144**  |
> | Phy.    | MCAR        | 0.689  | 0.659  | 0.466   | 0.739  | 0.544  | 0.617  | 0.444  | **0.343**  |
> |         | MAR         | 0.679  | 0.660  | 0.593   | 0.739  | 0.550  | 0.724  | 0.454  | **0.356**  |
> |         | MNAR        | 0.726  | 0.655  | 0.608   | 0.743  | 0.566  | 0.715  | 0.476  | **0.366**  |
>
> Combining the results in Table 1 and Fig.4, Fig.7 and Fig.8,  it can be found that the variations of CRPS are basically the same as RMSE and MAE for a specific model with different settings.
> Thanks again to the authors for their valuable suggestions. We will add this part of the experiment in the Appendix.

---

> ### Author Response · Authors · 2024-08-07
> **Continued response for W4**
>
> **4. (W4) Compared with the frequency domain method, and the method address the three terms**
>
> Thank you for the valuable suggestions.
> We add LaST [f] and FreTS [e] as baselines for comparison.
> Since they focus on the time series forecasting task,  we adapt them to the imputation task based on the TimesNet [48] setting.
> Furthermore, we have considered the TIDER model which takes into account trend, seasonal and residual terms in the MF process.
> We first report the part of updated Table 1 for the added baselines:
> | 　     | Miss. Rate | Metric |  LaST  |  FreTS |    FGTI    |
> |--------|------------|--------|:------:|:------:|:----------:|
> | KDD    | 10%        | RMSE   | 0.473  | 0.630  | **0.406** |
> |        | 　         | MAE    | 0.287  | 0.412  | **0.149** |
> |        | 20%        | RMSE   | 0.532  | 0.741  | **0.451** |
> |        | 　         | MAE    | 0.310  | 0.489  | **0.161** |
> |        | 30%        | RMSE   | 0.574  | 0.796  | **0.448** |
> |        | 　         | MAE    | 0.350  | 0.546  | **0.176** |
> |        | 40%        | RMSE   | 0.634  | 0.850  | **0.478** |
> | 　     | 　         | MAE    | 0.393  | 0.591  | **0.205** |
> | Guang. | 10%        | RMSE   | 0.347  | 0.456  | **0.230** |
> |        | 　         | MAE    | 0.244  | 0.340  | **0.170** |
> |        | 20%        | RMSE   | 0.440  | 0.602  | **0.258** |
> |        | 　         | MAE    | 0.312  | 0.460  | **0.176** |
> |        | 30%        | RMSE   | 0.545  | 0.709  | **0.291** |
> |        | 　         | MAE    | 0.388  | 0.547  | **0.202** |
> |        | 40%        | RMSE   | 0.637  | 0.787  | **0.356** |
> | 　     | 　         | MAE    | 0.458  | 0.611  | **0.254** |
> | Phy.   | 10%        | RMSE   | 0.768  | 0.804  | **0.580** |
> |        | 　         | MAE    | 0.516  | 0.540  | **0.286** |
> |        | 20%        | RMSE   | 0.786  | 0.825  | **0.577** |
> |        | 　         | MAE    | 0.550  | 0.576  | **0.309** |
> |        | 30%        | RMSE   | 0.825  | 0.861  | **0.624** |
> |        | 　         | MAE    | 0.578  | 0.603  | **0.336** |
> |        | 40%        | RMSE   | 0.850  | 0.883  | **0.669** |
> | 　     | 　         | MAE    | 0.603  | 0.626  | **0.376** |
>
> We then report the results for varying missing mechanisms with 10% missing values in the following table, and update Fig.7, Fig.8 and Fig.9:
> | 　     | Miss. Mech. | Metric | LaST   | FreTS  | FGTI       |
> |--------|-------------|--------|--------|--------|------------|
> | KDD    | MCAR        | RMSE   | 0.473  | 0.630  | **0.406** |
> |        | 　          | MAE    | 0.287  | 0.412  | **0.149** |
> |        | MAR         | RMSE   | 0.473  | 0.630  | **0.406** |
> |        | 　          | MAE    | 0.287  | 0.412  | **0.149** |
> |        | MNAR        | RMSE   | 0.619  | 0.809  | **0.499** |
> |        | 　          | MAE    | 0.326  | 0.473  | **0.174** |
> | Guang. | MCAR        | RMSE   | 0.347  | 0.456  | **0.230** |
> |        | 　          | MAE    | 0.244  | 0.340  | **0.170** |
> |        | MAR         | RMSE   | 0.327  | 0.465  | **0.218** |
> |        | 　          | MAE    | 0.234  | 0.356  | **0.150** |
> |        | MNAR        | RMSE   | 0.309  | 0.441  | **0.200** |
> |        | 　          | MAE    | 0.227  | 0.337  | **0.140** |
> | Phy.   | MCAR        | RMSE   | 0.768  | 0.804  | **0.580** |
> |        | 　          | MAE    | 0.516  | 0.540  | **0.286** |
> |        | MAR         | RMSE   | 0.309  | 0.441  | **0.200** |
> |        | 　          | MAE    | 0.227  | 0.337  | **0.140** |
> |        | MNAR        | RMSE   | 0.768  | 0.804  | **0.580** |
> |        | 　          | MAE    | 0.516  | 0.540  | **0.286** |
>
> After this, we report the resource consumption over KDD dataset with 10% missing values in the following table,
> and update Fig.5:
> |          | GPU Consumption Usage (MiB) | Running Time (s) |
> |----------|-----------------------------|------------------|
> | LaST     | 444                         | 4.63             |
> | FreTS    | 1118                        | 9.09             |
>
> Finally we report the updated downstream application experiments in the following table, and update Fig.6:
> |            | Air quality prediction (RMSE) | Mortality forecast (AUC) |
> |------------|-------------------------------|--------------------------|
> | LaST      | 0.64                          | 0.80                     |
> | FreTS     | 0.65                          | 0.80                     |
> | FGTI       | **0.59**                      | **0.86**                 |
>
> We can find that our FGTI still outperforms all baselines due to the guidance of high-frequency information and dominant-frequency information, as well as the generating ability of the diffusion model.

---

> ### Author Response · Authors · 2024-08-07
> **Continued response for W5, L1**
>
> **5. (W5) What cause lower MAR and MNAR improvements compared to MCAR for Guangzhou and PhysioNet**
>
> Thank you for your question.
> MCAR, MAR and MNAR reflect different missing scenarios in reality.
> For MCAR, the missing value is not dependent on other attributes.
> For MAR, the probability of missingness depends only on available information (e.g. missing records of traffic flow data may be related to rush hour, activities in public places).
> For MNAR, the missing observation depends on othe the attribute itself (e.g. reliability of recording devices).
> The effect of the models on different mechanisms depends on the temporal and attribute relations they capured in the dataset.
> Under the MAR and MNAR mechanisms, the critical temporal or attribute dependencies are more likely to be missing compared to MCAR, and therefore models may perform not well for MAR and MNAR.
> This also reflects the importance of introducing frequency domain information to guide the imputation.
>
> **6.  (L1) The contribution of the proposed method might be incremental**
>
> Thank you for your comments.
> **As far as we know, this is the first work to recognize the insight of paying special attention to residual term when impute time series, and introduce frequency-domain information to guide generative models.**
> We choose to implement FGTI through the diffusion model because diffusion models are currently demonstrating excellent performance in several fields [17,23].
> The frequency domain filters and the cross-domain representation learning module can be flexibly ported to other generative models.
> We think this insight can shed light into potential future directions in time series imputation.
> Therefore, we argue that our FGTI is not an incremental work.
>
> References:
>
> [a] MSTL: A Seasonal-Trend Decomposition Algorithm for Time Series with Multiple Seasonal Patterns
>
> [b] A wavelet-based approach for imputation in nonstationary multivariate time series
>
> [c] Rethinking general time series analysis from a frequency domain perspective
>
> [d] FEDformer: Frequency Enhanced Decomposed Transformer for Long-term Series Forecasting
>
> [e] Frequency-domain MLPs are More Effective Learners in Time Series Forecasting
>
> [f] Learning Latent Seasonal-Trend Representations for Time Series Forecasting

---

### Author Rebuttal · Authors · 2024-08-07

We sincerely thank all the reviewers for their thoughtful and constructive comments. We are greatly encouraged that they found our idea and contributions to be significant (Reviewer WWjA, qcDA, Q2t8 and SUMs), and technical sound (Reviewer WWjA, qcDA, SUMs). We are grateful that they identified our method to be effective (Reviewer WWjA, qcDA, Q2t8 and SUMs) and our paper to be well-written (Reviewer WWjA, qcDA, Q2t8). However, we believe that there are still several questions and concerns mentioned in the reviews that need to be addressed.
Meanwhile, we also revised the paper and the appendix according to the reviewers' valuable suggestions. The main changes are as follows:
- We add several competitive imputation methods (i.e. LaST, FreTS, TimeCIB) recommended by reviewer WWjA and qcDA in our experiments.
- Thanks to the comments from WWjA and SUMs，we add a section about experimental results of CRPS metric as suggested by reviewer .
- We also add a case study exploring the role of high-frequency condition and dominant-frequency condition in the Appendix, relying on the comments of Reviewers WWjA and qcDA.
- We provide the experiment analysis of the mask ratio and mask pattern in the Appendix as recommended by Reviewer qcDA.
- We check the full manuscript thoroughly and improve the presentation and grammar as recommended by Reviewer qcDA and Q2t8.
- As shown in the attached pdf file, we update Fig. 1 according to Reviewer qcDA's suggestion and add a figure presenting the detailed architecture of the denoising network according to Reviewer Q2t8's comment.

Best,

Authors

---

### Decision · Program_Chairs · 2024-09-25

**Decision:**

Accept (poster)

**Comment:**

This paper proposes a method called Frequency-aware Generative Models for Multivariate Time Series Imputation (FGTI). The approach aims to more carefully deal with high-frequency residual components when performing time series imputation tasks. The paper has uniformly positive reviews. The reviewers indicate that the paper is generally well-written and well-motivated. The experiments are extensive with a large and representative set of prior approaches and strong results. Some additional baselines and ablations were suggested and the authors provided them during the rebuttal phase. The reviewers who checked in during the author response indicated that their concerns were addressed by the author response.